# 𝕏olver: Generalist Reasoning and Problem Solving through Federated Multi-Agent Dynamics and Holistic Experience Learning

## Abstract

Despite rapid advances in complex reasoning, large language models (LLMs) largely operate in isolation, treating each problem as a fresh attempt without retaining or reusing prior experience. In contrast, expert problem solvers—such as Olympiad or programming contest teams—leverage a rich tapestry of experience: mentorship from coaches, intuition from past problems, mastery of tools and libraries, peer strategies, and continuous refinement through trial and error—even drawing insights from related problems under competition conditions. Inspired by this, we introduce 𝕏**olver**—a training-free, generalist reasoning and problem solving framework that equips a black-box LLM with a persistent, evolving memory of holistic experience. 𝕏olver combines two key innovations: (i) a holistic experience-learning paradigm that unifies external and self-retrieval, tool use, collaborative agent interaction, agent-driven evaluation, and iterative reasoning refinement; and (ii) a dynamic multi-agent collaboration schema that departs from orchestration engineering, instead employing a federated learning strategy in which agents independently solve problems and aggregate their solutions. Extensive evaluations across reasoning, agentic, and coding benchmarks show that 𝕏olver consistently outperforms specialized reasoning agents (e.g., OctoTools, Search-o1, AWorld, OpenHands, OAgents, Agent S2.5). Even with lightweight backbones (e.g., QWQ-32B), it frequently surpasses state-of-the-art proprietary models (Qwen3-235B, Gemini 2.5 Pro, o3, Deep Research, o4-mini-high). With stronger backbones (e.g., o3-mini-high), 𝕏olver achieves new state-of-the-art scores: 94.4 on AIME'24, 93.7 on AIME'25, 91.6 on LiveCodeBench, 90.1 on GAIA, 71.7 on BrowseComp, 74.4 on OSWorld, 84.9 on SWE-bench Verified (bash only), 57.3 on HLE, 94.6 on GPQA Diamond, 84.4 on 2WIKI, and 82.3 on Bamboogle—highlighting holistic and federated experience learning as a crucial step toward dynamic, generalist agents capable of expert-level reasoning. We will open-source all code, and data.

## 1 Introduction

Recent advances in large language models (LLMs) have made remarkable progress in complex reasoning and problem solving across domains including mathematics (Cobbe et al., 2021; Hendrycks et al., 2021; Lewkowycz et al., 2022) and programming (Chen et al., 2021; Austin et al., 2021; Khan et al., 2024). Yet despite these impressive capabilities, conventional LLM reasoning approaches remain fundamentally limited: they standalone each problem instance, generating solutions from scratch without accumulating or transferring insights from rich, diverse experiential knowledge.

This isolated reasoning paradigm marks a significant departure from how expert human problem solvers operate. Expert problem solvers—such as an Olympiad or programming contest teams—rarely approach problems in a vacuum. Instead, they draw upon a rich tapestry of cumulative experiences: absorbing mentorship from coaches, developing intuition from past problems, leveraging knowledge of tool usage and library functionality (e.g., calculator), adapting strategies based on peers' expertise and experiences, gaining insights through iterative trial and error, and learning from related problems even during competition. This holistic experience empowers them to tackle new challenges not from scratch, but by dynamically applying accumulated knowledge and adaptive strategies.

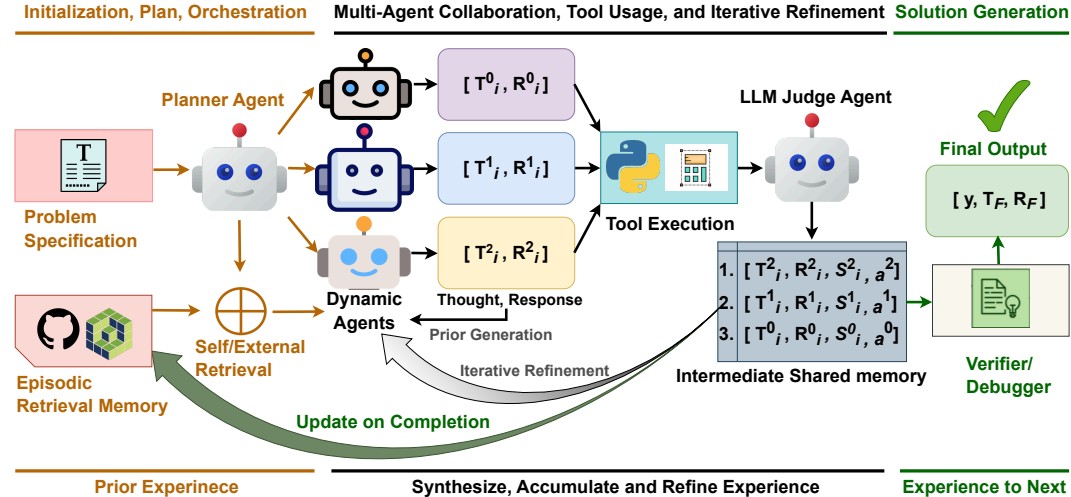

Figure 1: 𝕏**olver Scaffold.** At each iteration, agents receive their past reasoning history and top-ranked exemplars to generate new thoughts and responses, using tools (e.g., code) as needed. A judge model ranks outputs, and an intermediate memory maintains the best responses over time. Exemplars are initialized via episodic retrieval and continually updated with high-quality solutions from the memory. Iteration stops when convergence or max steps are reached, followed by final verification.

While numerous prior studies have enhanced LLM reasoning and problem solving through various forms of experiential knowledge augmentation, they have predominantly operated within discrete modalities—retrieving similar problems or relevant contexts (Parvez et al., 2021; Lewis et al., 2020; Guu et al., 2020), leveraging external tools (Lu et al., 2025; 2024), or facilitating multi-agent collaboration (Islam et al., 2025). Despite their individual strengths, these approaches address distinct facets of experiential knowledge independently, preventing LLMs from accumulating and synthesizing a comprehensive repertoire of learning signals across diverse experiential dimensions, thereby limiting the development of the rich, interconnected knowledge structures that characterize human expertise. This fragmentation is further compounded by the fact that most problem-solving frameworks are designed for specific tasks or environments, limiting their generality and the ability to transfer knowledge across domains.

In this paper, we introduce 𝕏olver—a training-free, generalist reasoning and problem solving framework that emulates the holistic experience-driven, collaborative reasoning of expert teams. At its core, 𝕏olver integrates planning, episodic retrieval from external and parametric memory, tool use, multi-agent collaboration, agent evaluation, and iterative self-refinement within a unified architecture. A planner agent coordinates specialized agents—such as mathematicians or programmers—that collaborate on complex tasks. Initially, reasoning is guided by exemplars from episodic memory; in subsequent iterations, agents rely on a shared memory that records trajectories, solutions, and feedback. This evolving memory enables agents to build on prior successes, correct errors, and progressively refine solutions. When necessary, agents invoke external tools (e.g., code execution), while a judge agent evaluates outputs, ranks top responses, and enriches the shared memory with curated traces. Iterations proceed until convergence or a fixed budget is reached, followed by a final verification step. Additionally, by updating its episodic store with each newly solved problem and its reasoning trace, 𝕏olver can continually expand its knowledge base. Figure 1 illustrates the workflow.

In addition to its holistic experience-driven paradigm, 𝕏olver differs fundamentally from existing multi-agent workflows, which often reduce to orchestration engineering over cascaded agent communication (Yao et al., 2022; Hong et al., 2024; Wang et al., 2024; Lei et al., 2024; Wu et al., 2023). In contrast, we introduce federated agent learning, a collaboration method in which each agent independently solves the complete problem without peer-to-peer messaging (federated), while a judge agent consolidates their reasoning, outputs, and evaluation feedback into a shared memory for that iteration (aggregated). Agents then condition on this evolving memory to iteratively refine their solutions. Finally, unlike task-specific frameworks, 𝕏olver 's human-aware design enables it to operate as a generalist problem solver.

We conduct large-scale experiments across a range of math and programming benchmarks—including GSM8K, Math-500, AIME (2024 and 2025), LiveCodeBench (v5), GAIA, BrowseComp, OSWorld,

Humanities Last Exams (HLE), SWE-bench Verified (bash only), GPQA Diamond, SQuAD2.0, 2WIKI and Bamboogle—using both proprietary (o3-mini and o4-mini) and open-weight (QWQ-32B and QVQ-72B-preview) backbone models. $\mathbb{X}$**olver** consistently outperforms specialized reasoning systems such as OctoTools, Search-o1, OpenHands, Agentless, WebSailor, OAgents, Alita, AWorld, GTA1, and Agent S2.5. Remarkably, even when instantiated with lightweight models such as QWQ-32 or o3-mini-med, $\mathbb{X}$olver often outperforms substantially larger state-of-the-art LLMs, including Qwen3-235B, Gemini 2.5 Pro, o3, and o4-mini-high. Moreover, with stronger backbones such as o3-mini-high or o4-mini-high $\mathbb{X}$olver establishes new state-of-the-art results across all benchmarks. Finally, our comprehensive ablations and analyses reveal how $\mathbb{X}$olver 's experiential components shape its performance, highlighting both its strengths and limitations.

## 2 THE $\mathbb{X}$OLVER FRAMEWORK

Given a problem query $q \in \mathcal{Q}$ and a pretrained language model $\text{LLM}_\theta(\cdot)$, a conventional approach generates a solution via single-step inference: $y \sim \text{LLM}_\theta(q)$. In contrast, $\mathbb{X}$**olver** executes a dynamic, multi-agent reasoning process that iteratively accumulates and leverages symbolic experience to solve complex problems more effectively. To support structured collaborative reasoning, $\mathbb{X}$olver maintains two complementary forms of memory: an *episodic memory* $\mathcal{D}_E$, which stores a library of past problems, solutions, and reasoning traces; and an intermediate dynamic *shared memory* $\mathcal{D}_S$, which evolves during inference to retain high-quality agent trajectories—comprising reasoning thoughts, responses, agent metadata, and feedback. In $\mathbb{X}$olver, a multi-agent team $\mathcal{A}$ is orchestrated adaptively by a planner agent $\mathcal{P}$, which assigns roles and configures memory access. During inference, $\mathcal{A}$ agents leverage an external toolset $\mathcal{T}$ (e.g., Python interpreter) to support accurate computation. Finally, a verifier or external debugger $\mathcal{V}$ is invoked to extract and format the final answer, and to validate correctness for executable outputs. Below, we first describe the $\mathbb{X}$olver agents and tools in Section 2.1, followed by the memory components in Section 2.2, and the inference cycle in Section 2.3.

### 2.1 AGENTS AND TOOLS

**Planner Agent** $\mathcal{P}$**.** The planner agent $\mathcal{P}$ is responsible for initiating, planning, and orchestrating the $\mathbb{X}$olver multi-agent architecture. Given the problem $q$ and the number of agents $m$, it constructs a team $\mathcal{A}$ of $m$ dynamic agents, each assigned a distinct expert role (e.g., algebra solver, mathematician, theorist, programmer, algorithm designer) tailored to the demands of $q$. To ensure sufficient task coverage and role diversity, $\mathcal{P}$ first prompts the underlying LLM to over-generate $M > m$ candidate agents, from which it then selects the most effective subset $\mathcal{A} \subset \{a_1, \dots, a_M\}$ such that $|\mathcal{A}| = m$. A summary of the most frequently generated and selected roles is provided in Appendix E.12.

**Dynamic Reasoning Agents** $\mathcal{A}$**.** The set $\mathcal{A} = \{a^1, a^2, \dots, a^m\}$ represents a team of dynamic reasoning agents constructed by the planner agent $\mathcal{P}$. Each agent $a^j \in \mathcal{A}$ is assigned a distinct expert role (e.g., algebra solver, programmer, counter-example generator) tailored to the task query $q$. Agents are instantiated using a standardized prompting template (see Appendix B) that incorporates the task description, assigned role, retrieved examples, prior reasoning attempts, and shared memory feedback—enabling iterative self-correction and role specialization.

At each iteration $i$, agent $a^j$ receives a context $\mathcal{C}_i^j$ and generates a structured reasoning trace $T_i^j$ and a response $R_i^j$. For the first iteration ($i = 0$), the context is initialized using the task query and relevant retrieved exemplars:

$$\mathcal{A} \leftarrow \mathcal{C}_0^j = \{q\} \cup \mathcal{R}(\mathcal{D}_E). \qquad \text{(BUILDCONTEXT)}$$

For subsequent iterations ($i \geq 1$), the context evolves by incorporating its prior generation (history) and the shared memory:

$$\mathcal{A} \leftarrow \mathcal{C}_i^j = \{q\} \cup \{T_{i-1}^j, R_{i-1}^j\} \cup \mathcal{D}_S. \qquad \text{(BUILDCONTEXT)}$$

**Judge Agent** $\mathcal{J}$**.** The judge agent $\mathcal{J}$ evaluates intermediate outputs from each agent and returns structured feedback to guide refinement and memory updates. Given a query $q$, a reasoning trace $T$, and a response $R$, it produces a feedback tuple $S = (T_S, s)$, where $T_S$ is a natural language explanation (e.g., critique, justification, correction), and $s$ is a scalar quality score. The interpretation of $s$ is task-dependent: for math and other tasks, $s \in [0, 1]$ reflects an LLM-estimated correctness

probability; for code tasks, $s \in \{0, 1, \ldots, N_{\text{test}}\}$, where $N_{\text{test}}$ denotes the total number of test cases including problem-provided samples and 10 synthesized test cases (LiveCodeBench) generated using AceCode-RM-32B (Zeng et al., 2025). This structured feedback enables agents to identify failures, receive localized corrections, and improve reasoning over iterations.

**Verifier Agent $\mathcal{V}$.** Due to linguistic complexity and varying answer specification formats, a response may be incorrect even when the underlying reasoning or open-ended response is valid. For instance, answer formats may require multiple-choice letters (e.g., "(A)" or "Choice B"). Such challenges are even more pronounced in code generation tasks, where predicted code may fail to execute or pass any sample test cases. To mitigate this, $\mathbb{X}$olver includes a Verifier Agent $\mathcal{V}$, which operates differently based on the output type. For math and other problems, $\mathcal{V}$ extracts the final reasoning $T_F$, response $R_F$, and answer $y$ from the response associated with the top-ranked entry BESTRESPONSE in $\mathcal{D}_S$, ensuring adherence to the expected output format. For executable code, $\mathbb{X}$olver invokes an external debugger (e.g., LDB (Zhong et al., 2024)), where $\mathcal{V}$ interacts with a Python runtime to capture execution feedback and iteratively fix runtime errors.

**Tools $\mathcal{T}$.** Integrating natural language reasoning with tools like Python execution is a proven way to boost performance on complex reasoning tasks Moshkov et al. (2025); Toshniwal et al. (2024). We observe that even advanced reasoning models often make mistakes in intermediate steps, particularly when computations become non-trivial. To address this, each dynamic agent $a^j$ is explicitly instructed to use Python execution during reasoning when needed. While for math and code tasks, $\mathbb{X}$olver limits $\mathcal{T}$ to Python (e.g., numerical libraries, interpreters, debuggers), our prompting strategy is tool-agnostic, allowed us an interface for extensions to richer toolsets such as (e.g., search engines, patch zoomer, clicking, typing, scrolling web pages, image captioner, file handler, knowledge databases, port transferer, reverse engineer) for other tasks. All prompts are 0-shot and provided in Appendix B.

## 2.2 MEMORY COMPONENTS

**Episodic Memory $\mathcal{D}_E$.** $\mathbb{X}$olver maintains two forms of episodic (long-term) memory: (1) an external memory corpus $\mathcal{D}_E^{\text{ext}} = \{(q', T', R')\}$, which consists of past problem instances $q'$, their corresponding reasoning traces $T'$ (optional), and solution responses $R'$; and (2) the internal parametric memory encoded in the weights of the agent-specific language model $\text{LLM}_j$.

We define a general retrieval operator $\mathcal{R}(\mathcal{D}_E)$ that returns a set of $K$ examples relevant to the query $q$. When $\mathcal{D}_E^{\text{ext}}$ is available, retrieval is conducted using similarity-based search (e.g., BM25):

$$\mathcal{R}(\mathcal{D}_E) = \{(q_k', T_k', R_k')\}_{k=1}^K \leftarrow \text{Retrieve}_j(q, \mathcal{D}_E^{\text{ext}}).$$

Otherwise, $\mathbb{X}$olver falls back to internal self-retrieval by sampling from the agent model itself:

$$\mathcal{R}(\mathcal{D}_E) = \{(q_k', T_k', R_k')\}_{k=1}^K \sim \text{LLM}_j(q).$$

In the case of an external episodic memory, $\mathcal{D}_E$ can also be updated with UPDATEEPISODICMEMORY by adding the top-ranked reasoning and response from $\mathcal{D}_S$, paired with the problem $q$, into the external corpus $\mathcal{D}_E^{\text{ext}}$. That is, $\mathcal{D}_E^{\text{ext}} \leftarrow \mathcal{D}_E^{\text{ext}} \cup (q, T, R)$, where $(T, R, S, a)$ is the top-ranked entry in $\mathcal{D}_S$.

**Intermediate Shared Memory $\mathcal{D}_S$.** The shared memory $\mathcal{D}_S$ maintains a fixed-size set of high-quality intermediate reasoning, responses, and metadata generated by the dynamic agents during inference on the current query $q$. For simplicity and to preserve the dynamic nature of the framework, we constrain $|\mathcal{D}_S| = m$, where $m$ is the number of dynamic agents in $\mathcal{A}$. Initially, $\mathcal{D}_S \leftarrow \emptyset$. At each iteration $i$, each agent $a_j \in \mathcal{A}$ produces a reasoning trace $T_i^j$, response $R_i^j$, and receives structured feedback $S_i^j = (T_S^{(i,j)}, s_{i,j})$ from the judge agent $\mathcal{J}$, where $T_S^{(i,j)}$ is a natural language explanation and $s_{i,j}$ is a scalar score reflecting the quality of the tuple $(T_i^j, R_i^j)$. After collecting the new outputs

$$\tau_i^j = (T_i^j, R_i^j, S_i^j, a^j), \quad j = 1, \ldots, m, \quad \text{(RUNAGENTS)}$$

we form the candidate pool $\mathcal{M} = \mathcal{D}_S \cup \{\tau_i^1, \ldots, \tau_i^m\}$. We then update the fixed-size shared memory by keeping only the top-$m$ tuples by score

$$\mathcal{D}_S \leftarrow \text{TopK}\big(\mathcal{M}, \ m; \ \text{key}(e) = s(e)\big), , \quad \text{(UPDATESHAREDMEMORY)}$$

where $s(e)$ extracts the scalar score from $e = (T, R, (T_S, s), a)$.

This replacement mechanism ensures that $\mathcal{D}_S$ always contains exactly $m$ entries with the highest observed scores across all iterations. By maintaining only the strongest reasoning-response-feedback tuples, the shared memory facilitates knowledge transfer between agents and across iterations, enabling collaborative improvement through exposure to diverse high-quality solutions.

### 2.3 INFERENCE PROTOCOL

Algorithm 1 summarizes the $\mathbb{X}$olver inference protocol, which operates in three structured stages. **Stage-1**, which emulates initialization with prior experience, involves the planner constructing a team of agents $\mathcal{A}$ (lines 2–3). **Stage-2**, embodying symbolic *federated* experience accumulation and refinement, iterates for $\mathcal{I}$ rounds (lines 4–10). In each round, all agents receive access to $\mathcal{D}_S$ and $\mathcal{D}_E$, build their symbolic *aggregated* contexts, and generate structured trajectories and responses. ($\mathcal{D}_E$ is only used for context construction at the first iteration). These are evaluated by the judge agent $\mathcal{J}$, and $\mathcal{D}_S$ is updated with the resulting feedback tuples (line 7).

---

**Algorithm 1** $\mathbb{X}$olver Inference Protocol

1: **Input:** Query $q$, Tools $\mathcal{T}$, Episodic Memory $\mathcal{D}_E$, parameters $m, k, I$
2: **Init:** $\mathcal{D}_S \leftarrow \emptyset$
3: $\mathcal{A} \leftarrow \text{PLANNER}(q, m)$
4: **for** $i = 0$ **to** $\mathcal{I}$ **do**
5:      $\{\mathcal{C}_i\}_{c=1}^m \leftarrow \text{BUILDCONTEXT}(\mathcal{A}, \mathcal{D}_E, \mathcal{D}_S, q, i)$
6:      $\{\tau_i^j\}_{j=1}^m \leftarrow \text{RUNAGENTS}(\mathcal{A}, \mathcal{C}_i, \mathcal{T}, \mathcal{J})$
7:      $\mathcal{D}_S \leftarrow \text{UPDATESHAREDMEMORY}(\mathcal{D}_S, \{\tau_i^j\})$
8:      **if** $\text{CONVERGED}(\mathcal{D}_S)$ **then**
9:          **break**
10:      **end if**
11: **end for**
12: $y \leftarrow \mathcal{V}(\text{BESTRESPONSE}(\mathcal{D}_S))$
13: $\text{UPDATEEPISODICMEMORY}(\mathcal{D}_E, q, \mathcal{D}_S)$
14: **Return** $y$

---

Upon convergence[1] or after $\mathcal{I}$ rounds, **Stage-3** invokes the verifier agent $\mathcal{V}$, which extracts the final answer from the top-ranked entry in $\mathcal{D}_S$ (line 11), and updates $\mathcal{D}_E$ with the new experience.

## 3 EXPERIMENTS

### 3.1 EVALUATION SETUP

**Evaluation Benchmarks.** We evaluate $\mathbb{X}$olver on math, coding, open-ended, multimodal, browsing, multihop, and agentic tasks. Math: GSM8K (Cobbe et al., 2021), Math-500 (Hendrycks et al., 2021), AIME 2024 (MAA, 2024), and AIME 2025 (MAA, 2025) (multi-step high-school competition problems). Coding: SWE-bench Verified (bash only) (Jimenez et al., 2023) and LiveCodeBench v5 (Jain et al., 2024) (dynamic, no-leakage). Open-ended: GPQA Diamond (Rein et al., 2024), SQuAD2.0 (Rajpurkar et al., 2018). Multimodal: GAIA (Mialon et al., 2023), HLE (Phan et al., 2025). Tool-integrated browsing: BrowseComp (Wei et al., 2025). Multihop: 2WIKI (Ho et al., 2020), Bamboogle (Press et al., 2022). Agentic: OSWorld (Xie et al., 2024). These span arithmetic, algebra, number theory, geometry, combinatorics, and algorithmic problem solving.

**Baselines and Metrics.** We compare $\mathbb{X}$olver to: (a) proprietary reasoning models—Gemini 2.5 (Pro, Flash Think) (DeepMind, 2025), Grok-3 Beta Think/Mini (Beta) Think (xAI, 2025), Claude 3.7 Sonnet Think (Anthropic, 2025), o1 (OpenAI, 2024b), o3-mini/o3/o4-mini (OpenAI, 2025); (b) open-weight LLMs—Qwen3-235B (Qwen Team, 2024), QWQ-32B (Qwen Team, 2025), DeepSeek-R1 (DeepSeek-AI, 2025); (c) specialized models—AlphaOne (Zhang et al., 2025), OpenMath-Reason (Moshkov et al., 2025), rStar-Math (Guan et al., 2025), rStar-Coder (Liu et al., 2025), OpenCodeReason (Ahmad et al., 2025), Kimi-K1-1.6 (Kimi Team, 2025), Kimi-k2-0905 (Team et al., 2025), Deep Research, ChatGPT Agent, and OpenCua-Qwen2-7B/32B (Wang et al., 2025); and (d) agent/framework baselines—Self-Reflexion (Shinn et al., 2023), Search-o1 (Li et al., 2025b), OctoTools (Lu et al., 2025), CheatSheet (Suzgun et al., 2025), CodeSim (Islam et al., 2025), Open-Hands (Wang et al., 2024), Agentless (Xia et al., 2024), WebSailor (Li et al., 2025a), OAgents (Zhu et al., 2025), Alita (Qiu et al., 2025), AWorld (Yu et al., 2025), GTA1 (Yang et al., 2025), and Agent S2.5 (Agashe et al., 2025). For (d) we re-implement with the same backbone LLMs as ours; for (a–c) we report official numbers. Metrics: accuracy for math and agentic tasks (judged by GPT-4o (OpenAI, 2024a)) and *pass@1* for code.

---

[1]$\mathbb{X}$olver converges when each dynamic agent in $\mathcal{A}$ holds the very same scores (same at max)—terminates when each agent is at max possible scores in the intermediate shared memory $\mathcal{D}_S$ or max iterations $\mathcal{I}$ elapsed.

**Inference Details.** Backbones include open-weight QWQ-32B (Qwen Team, 2024), QVQ-72B-preview, and proprietary o3-mini/o4-mini (medium/high) (OpenAI, 2025). To reduce variance, we report averages over 32 runs on LIVECODEBENCH and AIME '25, and 16 runs on AIME '24, with $\sim 1\%$ std. dev. (Appendix E.10). For simpler or instructional tasks—GSM8K, MATH-500, GAIA, BrowseComp, OSWorld, SWE-bench, HLE, and GPQA Diamond—we follow DeepSeek-v3 (Liu et al., 2024) with single greedy decoding. Defaults: temperature 0.2, agents $m = 3$, max iterations $\mathcal{I} = 2$; terminate on reaching $\mathcal{I}$ or when shared memory $\mathcal{D}_s$ converges (score 1.0 for non-code tasks; all tests passed for code). For external retrieval in coding (LiveCodeBench), we build a 9M-token corpus of algorithmic problems with C++ solutions from GitHub[2] (see Appendix D); for math we use OPENMATHREASON (Moshkov et al., 2025). Other tasks use self-retrieval (SWE-bench, SQuAD2.0, GPQA Diamond) or adaptive external retrieval (GAIA, OSWorld, HLE, 2WIKI, Bamboogle, BrowseComp) with search engines. We evaluate two variants of $\mathbb{X}$olver: (i) $\mathbb{X}$olver *with in-competition cross-problem experience* ($\mathbb{X}$olver (+)), which dynamically updates the episodic memory after solving each problem to utilize accumulated knowledge across problems; and (ii) $\mathbb{X}$*olver (−)*, which keeps the episodic memory static, focusing solely on problem-specific experience. By default, we refer to $\mathbb{X}$olver (+) as our method if not specified otherwise.

## 3.2 MAIN RESULTS

Table 1 evaluates $\mathbb{X}$olver across diverse agentic, reasoning, and coding tasks, compared to state-of-the-art LLMs, specialized models, and other frameworks. Whereas, Table 2 (Appendix E.1) presents results in more open-ended and multi-hop reasoning tasks, showing its effectiveness and generality.

**Strong Gains Across Benchmarks.** Across all datasets, both $\mathbb{X}$olver (−) and $\mathbb{X}$olver (+) surpass the backbone LongCoT baselines: for o3-mini-medium, AIME'24 improves 75.8→93.8 and LiveCodeBench 66.3→87.3; for o4-mini-medium, GAIA 54.7→85.6 and HLE 18.1→49.0; for QWQ-32B, AIME'24 78.1→93.6 and LiveCodeBench 63.4→79.2; and for QVQ-72B-preview, GAIA 45.2→77.3 and HLE 11.7→38.5. **Surpassing Prior Agents.** Relative to Search-o1, OctoTools, CheatSheet, Agent S2.5, and AWorld, $\mathbb{X}$olver (+) delivers consistent gains, exceeding the best baseline by +12.7 on AIME'25 and +13.5 on LiveCodeBench with o3-mini-medium, and by +14.8 on OSWorld and +9.8 on GAIA with o4-mini-medium. **In Comparison to Leading LLMs.** Despite weaker backbones, $\mathbb{X}$olver (+) matches or surpasses frontier models, outperforming o4-mini-high with o3-mini-medium on AIME'24 (93.8 vs. 93.4) and substantially on LiveCodeBench (87.3 vs. 69.5). **Backbone Agnostic** The framework yields consistent benefits across backbones, as $\mathbb{X}$olver (+) attains 97.1 on GSM8K with o3-mini-medium and 98.0 with QWQ-32B, both markedly above their respective baselines. **Effectiveness of Dynamic Episodic Memory.** The cross-problem variant $\mathbb{X}$olver (+) reliably outperforms the problem-specific $\mathbb{X}$olver (−), averaging +4.8 points across backbones, with the largest gain of +8.2 on BrowseComp using o3-mini-medium. **Scales with Backbone LLM's Strength.** Performance scales with stronger backbones—as shown by o3-mini-high (GSM8K 98.1, AIME'24 94.4, AIME'25 93.7, Math-500 99.8, LiveCodeBench 91.6, BrowseComp 71.7, SWE-bench 84.9, GPQA 94.6) and o4-mini-high (GAIA 90.1, OSWorld 74.4, HLE 57.3)—and remains backbone-agnostic, with GSM8K reaching 97.1 on o3-mini-medium and 98.0 on QWQ-32B, collectively setting new state-of-the-art results across all benchmarks.

## 4 ABLATION AND ANALYSES

**Ablation-Studies:**

In Figure 2, we present an ablation study quantifying the contribution of individual components in $\mathbb{X}$**olver** to overall performance, measured by the average performance drop on math reasoning (Math Avg) and programming (LCB) tasks. Each component plays a necessary role, with the most significant degradation observed when removing Multi-iteration and

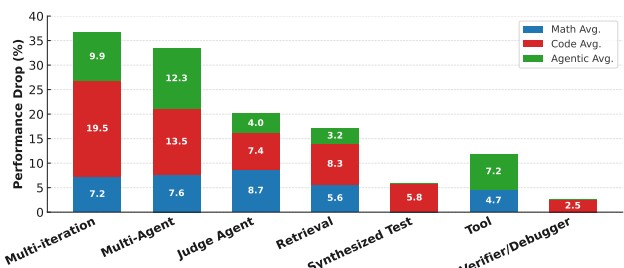

Figure 2: Performance drop when removing each component from $\mathbb{X}$olver. Bars show average drop on Math (bottom), Code (middle) and Agentic (top).

---

[2] https://github.com/cp-algorithms/cp-algorithms

| Model | Appr. | GSM 8K | AIME '24 | AIME '25 | Math -500 | LCB (v5) | GAIA | Browse Comp | OS World | SWE-V (bash) | HLE | GPQA Diam. |
|---|---|---|---|---|---|---|---|---|---|---|---|---|
| **Proprietary Models** | | | | | | | | | | | | |
| Claude 3.7 Sonnet T. | LongCoT | – | 61.3 | 49.5 | 96.2 | 51.4 | 53.0 | 4.8 | 35.6 | 52.8 | 8.0 | 75.3 |
| Grok-3 Beta T. | Direct | – | 83.9 | 77.3 | – | 70.6 | – | – | – | – | – | 73.7 |
| Gemini 2.5 F. T. | LongCoT | – | 88.0 | 78.0 | – | 63.5 | 12.1 | 79.0 | – | – | – | – |
| o1 | LongCoT | 96.4 | 74.3 | 79.2 | 96.4 | 71.0 | – | 9.9 | – | – | 8.0 | 73.0 |
| o3-mini-high | LongCoT | – | 87.3 | 86.5 | – | 69.5 | | | | | | |
| Gemini 2.5 Pro | Direct | – | 92.0 | 86.7 | – | 70.4 | – | – | – | 45.0 | 21.6 | 84.4 |
| o3 | LongCoT | 96.7 | 91.6 | 88.9 | – | – | – | 49.7 | 23.0 | 58.4 | 20.3 | 82.7 |
| o4-mini-high | LongCoT | – | 93.4 | **92.7** | – | 69.5 | – | 28.3 | – | – | 18.1 | 81.4 |
| **Open Weights Models** | | | | | | | | | | | | |
| DeepSeek-R1 | LongCoT | – | 79.8 | 70.0 | 97.3 | 64.3 | – | 9.5 | – | – | 8.5 | 81.3 |
| Qwen3-235B-A22B | LongCoT | – | 85.7 | 81.5 | – | 70.7 | – | – | – | – | 11.8 | 79.0 |
| **Specialized Models (with Best Reported Results)** | | | | | | | | | | | | |
| rStar-Math | | 95.2 | 53.3 | – | 90.0 | – | – | – | – | – | – | – |
| OpenMathReason | – | – | 93.3 | 80.0 | – | – | – | – | – | – | – | – |
| AlphaOne | – | – | 53.3 | – | 89.4 | 75.8 | – | – | – | – | – | – |
| OpenCodeReason | – | – | – | – | – | 61.8 | – | – | – | – | – | – |
| rStar-Coder | – | – | – | – | – | 62.5 | – | – | – | – | – | – |
| Kimi-k1.6-IOI-high | – | – | – | – | – | 73.8 | – | – | – | – | – | – |
| Kimi-k2-0905 | – | – | – | – | – | – | – | – | – | – | – | 76.7 |
| Deep Research | – | – | – | – | – | – | 63.7 | 51.5 | – | – | 26.6 | – |
| ChatGPT Agent | – | – | – | – | – | – | – | **68.9** | – | – | – | – |
| OpenCua-32B | – | – | – | – | – | – | – | – | – | 34.8 | – | – |
| **Reasoning Agents/Frameworks** | | | | | | | | | | | | |
| | LongCoT | 95.2 | 75.8 | 70.4 | 97.3 | 66.3 | 54.7* | 24.4 | 31.4* | 56.2 | 18.1* | 75.0 |
| | Self-Refl. | 93.1 | 79.4 | 76.5 | 95.2 | 73.2 | – | – | – | – | – | – |
| | OctoTools | 95.4 | 81.7 | 75.3 | 97.5 | – | – | – | – | – | – | – |
| | Search-o1 | 95.8 | 81.8 | 76.7 | 97.6 | 73.6 | – | – | – | – | 30.4* | 81.2 |
| | Alita | – | – | – | – | – | 74.1* | – | – | – | – | – |
| | AWorld | – | – | – | – | – | 75.8* | – | – | – | – | – |
| | GTA1 | – | – | – | – | – | – | – | 55.3* | – | – | – |
| o3-mini-medium | Agent S2.5 | – | – | – | – | – | – | – | 59.0* | – | – | – |
| | CheatSheet | 95.9 | 82.2 | 75.8 | 97.7 | – | – | – | – | – | – | – |
| | CodeSim | – | – | – | 73.8 | – | – | – | – | – | – | – |
| | OpenHands | - | – | – | – | – | – | – | – | 62.4 | – | – |
| | Agentless | – | – | – | – | – | – | – | – | 70.0 | – | – |
| | WebSailor | – | – | – | – | – | – | 45.5 | – | – | – | – |
| | OAgents | – | – | – | – | – | – | 47.1 | – | – | – | – |
| | 𝕏olver(−) | 95.6 | 87.2 | 85.1 | _97.7_ | 79.6 | _82.4‡_ | 60.7 | 69.9* | 73.9 | _42.6‡_ | 85.4 |
| | 𝕏olver(+) | _97.1_ | 93.8 | _89.4_ | 99.2 | 87.3 | 85.6‡ | 68.9 | 73.8* | 81.5 | 49.0‡ | 89.1 |
| | LongCoT | 96.1 | 78.1 | 65.8 | 83.2 | 63.4 | 45.2* | 17.6 | 14.1* | 30.1 | 11.7* | 58.1 |
| | Self-Refl. | 94.0 | 79.3 | 66.3 | 80.4 | 69.2 | – | – | – | – | – | – |
| | OctoTools | 96.3 | 83.0 | 71.7 | 86.1 | – | – | – | – | – | – | – |
| | Search-o1 | 96.4 | 84.4 | 71.8 | 87.1 | 69.3 | – | – | – | – | 21.2* | 63.6 |
| | Alita | – | – | – | – | – | 66.3* | – | – | – | – | – |
| | AWorld | – | – | – | – | – | 67.3* | – | – | – | – | – |
| | GTA1 | – | – | – | – | – | – | – | 37.1* | – | – | – |
| QWQ-32B | Agent S2.5 | – | – | – | – | – | – | – | 39.2* | – | – | – |
| | CheatSheet | 96.8 | 83.5 | 72.2 | 86.5 | – | – | – | – | – | – | – |
| | CodeSim | – | – | – | – | 70.5 | – | – | – | – | – | – |
| | OpenHands | - | – | – | – | – | – | – | – | 50.7 | – | – |
| | Agentless | – | – | – | – | – | – | – | – | 57.4 | – | – |
| | WebSailor | – | – | – | – | – | – | 37.0 | – | – | – | – |
| | OAgents | – | – | – | – | – | – | 38.5 | – | – | – | – |
| | 𝕏olver(−) | 96.5 | 89.9 | 79.5 | 93.1 | 76.2 | 75.6‡ | 49.3 | 46.4* | 68.2 | 30.9‡ | 70.1 |
| | 𝕏olver(+) | **98.0** | _93.6_ | 82.7 | 95.5 | 79.2 | 77.3‡ | 57.4 | 50.9* | 71.6 | 38.5‡ | 75.4 |
| o3-mini-high | 𝕏olver (+) | **98.1** | **94.4** | **93.7** | **99.8** | **91.6** | **90.1‡** | **71.7** | **74.4*** | **84.9** | **57.3‡** | **94.6** |

Table 1: Comparison of 𝕏olver against SoTA reasoning models, specialized models, and other reasoning agents agentic, mathematical and coding tasks. Best results are boldfaced and second-best results are underlined. F: Flash, T: Think models, LongCoT: standard prompting for reasoning models, *: multimodal dataset, so we use `o4-mini` instead `o3-mini` and use `QVQ-72-preview` instead `QWQ-32B`. "-" denotes either n/a (e.g., only specialized models) or results not reported.

Multi-Agent followed by Judge Agent, highlighting their central importance. In contrast, removing Verifier/Debugger and Tool leads to comparatively smaller drops, suggesting a more auxiliary role in the overall system. Verifier is critical for Math tasks and cannot be removed, while Tool apply only to Math and Agentic tasks, whereas Test only for Code tasks. Likewise retrieval where self-retrieval can also work in-place of external retrieval with some drop in accuracy.

**Impact of Agent and Tool Count with Iterations, and Emerging Benefits of Collaboration.** Figure 3, performance improves consistently on both AIME '25 and LIVECODEBENCH with more agents or iterations. From Figure 3, it is evident that for coding tasks, increasing the number of iterations is more crucial than increasing the number of agents. However, for mathematical tasks, both agents and iterations are similarly important. To investigate further, we conducted a more granular experiment on the AIME '25 dataset. We found that, when keeping the total

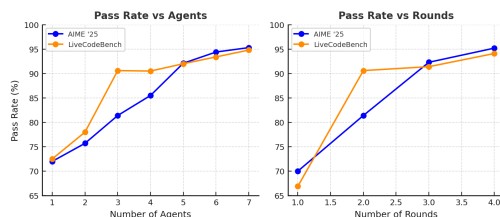

Figure 3: Impact of iterations and agents in $\mathbb{X}$olver on AIME '25 (QWQ-32B) and LIVE-CODEBENCH (o3-mini-medium).

budget (i.e., agents × iterations) constant, increasing the number of agents yields better performance—achieving over 4% improvement compared to increasing iterations. More details as well as the impact of tool count vs. iterations (see Figure 7) on $\mathbb{X}$olver performance are in Appendix E.3.

**Effect of Retrieval Strategies on $\mathbb{X}$olver Performance.** Figure 8 shows that, performance on both AIME '25 and LIVECODEBENCH follows the trend: *External Retrieval > Self-Retrieval > No Retrieval*, indicating that external retrieval significantly enhances $\mathbb{X}$olver's performance. Nevertheless, our dual-memory design gives flexibility about the dependence on retrieval corpora: while external retrieval gives the best results, self-retrieval from the model's own parametric memory or even no retrieval still gets SOTA results. More details can be found in Appendix E.4.

**Fine-grained Performance Analysis.** In Figure 4, we present fine-grained performance comparisons on GAIA dataset, evaluating $\mathbb{X}$olver against the agentic baseline framework AWorld. As observed, $\mathbb{X}$olver consistently outperforms AWorld across all difficulty levels using both QVQ-72B-preview and o4-mini-medium backbones. The performance gap is particularly notable with the open-weight QVQ-72B-preview model and on more challenging problems where AWorld struggles—such as

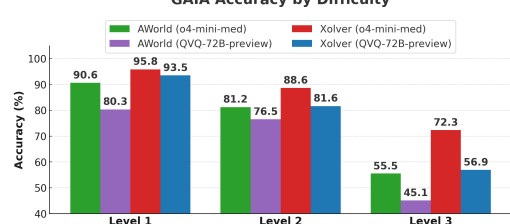

Figure 4: Performance comparison per difficulty levels in GAIA.

Level 3 problems in GAIA. More details and further fine-grained analysis on MATH-500 (see Figure 9), LCB (see Figure 10) and GPQA DIAMOND (see Figure 11) have shown in Appendix E.5.

**Can a Self-Judge Replace a Judge Agent?** As in self-reflection, each agent can also judge its own reasoning and response without changing its role. We observe that this does not compensate a different judge Agent, dropping average performance 8.7% in math, 4% in agentic and 7.4% in coding. Further details can be found at Appendix E.6.

**Cost Analysis and How Long Do $\mathbb{X}$olver Agents Think?** In Figure 12, we summarize input, reasoning, and output token usage for $\mathbb{X}$olver across all datasets. Token complexity scales as $O(m\mathcal{I})$ with $m$ agents and $\mathcal{I}$ iterations, while runtime remains $O(\mathcal{I})$ because agents run in parallel. This is more efficient than self-consistency (Wang et al., 2022), which typically requires 32–64 generations per example, and than the CheatSheet baseline, whose usefulness estimation induces $O(n^2)$ memory updates. As a multi-agent system, $\mathbb{X}$olver allocates roughly 25% of tokens to core reasoning, with the remainder spent on context sharing and inter-agent communication. Relative to the single-agent baseline Search-o1, $\mathbb{X}$olver consistently uses about 2× more tokens, a cost offset by substantial performance gains. More details can be found at Appendix E.7.

**Does Data Shuffling Affect $\mathbb{X}$olver Performance?** While we report without any shuffling, shuffling can change the results but minimally (see Table 4). Further details can be found at Appendix E.8.
**Qualitative Examples.** As in Appendix C, we present examples of full cycle $\mathbb{X}$olver output on both math and code reasoning tasks demonstrating a full view of how $\mathbb{X}$olver gradually accumulates.

**More Error Analysis.** We analyze errors across math, code, and agentic tasks beyond accuracy or pass@1. Although both backbone LLMs generate solution, that are not optimized and hence did not pass. More details as well as further error analysis on math and code (see Figure 13) and agentic tasks (see Figure 14) can be found at Appendix E.9.

**Dynamics of Reasoning Patterns in 𝕏olver Traces.** We study how 𝕏olver adapts its reasoning on LCB by tracking reasoning-pattern frequencies across difficulty tiers (see Table 5). Details of pattern collection are in Appendix E.10. As difficulty rises, 𝕏olver increases *self-evaluation* and *exploratory strategies* (e.g., trying new approaches). Correct solutions show reduced rephrasing and subgoal decomposition, indicating more direct, confident reasoning; incorrect ones exhibit more subgoal setup and rephrasing, reflecting failure recognition and restructuring. Unlike OPENCODEREASONING, which stagnates or regresses in key patterns (e.g., self-evaluation), 𝕏olver remains robust and adaptive, aided by multi-agent collaboration and judge feedback, highlighting its generality and flexibility.

**Case-Study: How 𝕏olver Enhances Reasoning**

To understand how 𝕏olver enhances performance, we conducted case studies on AIME '25 and LIVECODEBENCH. As shown in our ablation study, at each iteration, the Judge Agent ranks the responses, and in the following round, all agents attempt to match at least one of the previously top-ranked outputs. As shown in Figure 15 (see Appendix E.11), over time, this leads to improved outputs from the agents, increased agreement among them, and eventual convergence. Notably, removing the Judge Agent significantly degrades performance, as demonstrated in our first ablation.

## 5 RELATED WORK

This section presents the concise view of related works. Full description can be found in Appendix A.

**Memory-Augmented and Retrieval-Augmented LLMs.** Memory augmentation progressed from static retrieval (RAG, REALM) to dynamic self-reflection/scratch-space methods (Reflexion, MemGPT, Scratchpads) but remains single-agent and task-isolated, limiting cross-problem learning (Lewis et al., 2020; Guu et al., 2020; Shinn et al., 2024; Packer et al., 2023; Nye et al., 2021). 𝕏**olver** introduces a dual-memory design—episodic long-term plus dynamic intermediate—enabling specialized agents to build a persistent, shared knowledge base beyond cross-trial sharing and multi-source memory that still assume single agents (Zhao et al., 2023; Shinn et al., 2024; Yao et al., 2023c; Feng et al., 2025). **Multi-Agent Problem Solving.** Collaborative frameworks (CAMEL, AgentVerse, AutoGen, DyLAN), multi-agent code generation/problem solving, and debate improve specialization yet typically address isolated tasks without experience retention (Guo et al., 2024; Li et al., 2023; Chen et al., 2023; Wu et al., 2023; Liu et al., 2023; Islam et al., 2024; 2025; Du et al., 2023; Rahman et al., 2025; Subramaniam et al., 2025; Feng et al., 2025). 𝕏**olver** adds dual-memory, holistic experience integration, judge-mediated selection, and continual episodic expansion to yield experience-aware agents. **LLM Reasoning Enhancement Techniques.** Chain-of-Thought, Self-Consistency, and Tree of Thoughts broaden reasoning search yet remain single-pass; self-reflective (Reflexion) and retrieval-enhanced (CheatSheet, Search-o1) methods iterate or retrieve but stay single-agent (Wei et al., 2022; Wang et al.; Yao et al., 2023a; Shinn et al., 2024; Suzgun et al., 2025; Li et al., 2025b). 𝕏**olver** unifies these within a multi-agent loop where judge-mediated iterations and dual memories support cross-problem learning. **Tool-Augmented Reasoning.** Tool use evolved from single-tool browsing and code execution (WebGPT, PAL) to multi-tool orchestration (ReAct, Chameleon, OctoTools), yet execution is usually single-agent without persistent learning (Nakano et al., 2021; Gao et al., 2023; Yao et al., 2023b; Lu et al., 2024; 2025). 𝕏**olver** reframes tools as shared, memory-enriched assets: agents coordinate tool calls, share outcomes, and accumulate reusable strategies across problems.

## 6 CONCLUSION

We present 𝕏olver, an open-source multi-agent inference framework for experience-aware reasoning. It combines: (1) episodic retrieval from external or self memory; (2) a growing shared memory of reusable, high-quality reasoning traces; (3) tool invocation for complex computation; (4) coordinated multi-agent reasoning; (5) self-evaluation with iterative refinement; (6) verification or external debugging; and (7) cross-problem strategy transfer. Together, these enable adaptive, context-informed problem solving and continual improvement across tasks. Current limits are higher token usage and

dependence on backbone LLMs. To strengthen validity, we will add advanced external verifiers and structured guardrails, moving $\mathbb{X}$olver toward more efficient, robust, and human-expert-like reasoning.

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

## A    RELATED WORK

**Memory-Augmented and Retrieval-Augmented LLMs.** Memory-augmented language models have evolved from static retrieval systems like RAG (Lewis et al., 2020) and REALM (Guu et al., 2020) to dynamic approaches such as Reflexion (Shinn et al., 2024), MemGPT (Packer et al., 2023), and Scratchpads (Nye et al., 2021). However, these systems operate on isolated tasks, lack cross-problem experience accumulation, and employ single-agent architectures. 𝕏**olver** addresses these limitations through a novel dual-memory architecture combining episodic long-term memory with dynamic intermediate memory, enabling specialized agents to collectively build and refine experiential knowledge. While prior work has explored cross-trial information sharing (Zhao et al., 2023; Shinn et al., 2024) and multi-source memory integration (Yao et al., 2023c), these approaches remain confined to single-agent settings. Our framework creates a persistent knowledge base through multi-agent collaboration (Feng et al., 2025), allowing agents to accumulate expertise from solved problems and leverage collective experience for future tasks.

**Multi-Agent Problem Solving.** Multi-agent LLM systems address the limitations of single models by leveraging collaborative approaches for improved reliability and task specialization (Guo et al., 2024; Feng et al., 2025). From early frameworks like CAMEL (Li et al., 2023) with fixed role assignments, the field progressed to dynamic role adjustment in AgentVerse (Chen et al., 2023) and code execution in AutoGen (Wu et al., 2023). Recent advances include layered agent networks in DyLAN (Liu et al., 2023), multi-agent code generation and problem solving (Islam et al., 2024; 2025) and multi-agent debate frameworks (Du et al., 2023; Rahman et al., 2025; Subramaniam et al., 2025). While these systems demonstrate effective collaboration, they operate on isolated problems without cross-task experience accumulation. 𝕏**olver** introduces dual-memory architecture, holistic experience integration, judge-mediated selection, and continuous episodic corpus expansion—transforming single-problem solvers into experience-aware agents.

**LLM Reasoning Enhancement Techniques.** Various techniques have emerged to enhance LLM reasoning capabilities beyond standard prompting. Chain-of-Thought (Wei et al., 2022) introduced step-by-step reasoning, Self-Consistency (Wang et al.) explores multiple reasoning paths with majority voting, and Tree of Thoughts (Yao et al., 2023a) enables exploration of reasoning branches—yet all remain limited to single-pass generation. Self-reflective approaches like Reflexion (Shinn et al., 2024) enable iterative improvement but operate within single tasks, while retrieval-enhanced methods like CheatSheet (Suzgun et al., 2025) and Search-o1 (Li et al., 2025b) remain confined to single-agent architectures. These approaches share fundamental limitations: no cross-problem learning, no persistent memory, and no multi-agent collaboration. 𝕏**olver** unifies these enhancements within a multi-agent framework where agents collaboratively refine solutions through judge-mediated iterations and leverage dual memory systems for cross-problem learning.

**Tool-Augmented Reasoning.** Tool integration extends LLM capabilities beyond language processing. Early systems like WebGPT (Nakano et al., 2021) introduced single-tool integration, while PAL (Gao et al., 2023) enabled code execution for mathematical reasoning. Multi-tool frameworks evolved with ReAct (Yao et al., 2023b) interleaving reasoning with actions, Chameleon (Lu et al., 2024) composing multiple tools, and OctoTools (Lu et al., 2025) standardizing tool planning—yet all remain limited to single-agent execution without iterative refinement or cross-problem learning. 𝕏**olver** transforms tool use into a collaborative, memory-enriched ecosystem where agents collectively execute tools, share outcomes, and accumulate successful strategies across problems—creating an adaptive framework that evolves with experience.

## B    LISTS OF PROMPTS

This section provides the list of prompts for planning, dynamic, judge, verifier and reasoning segmentation we have used in the experimental period. These are crucial to ensure the reproducibility (Laskar et al., 2024) of the framework 𝕏**olver**.

## B.1 PLANNER AGENT

---

**Prompt for PLANNER AGENT**

You are a planner to solve a {coding/math/agentic} problem. Here is the problem for which you have to plan:
`{problem_dict[query['problem_id']]['description']}`

First draft required strictly greater than `{m}` specialized roles to solve the problem collaboratively with reasoning behind your draft of each role.

Then select the highly influential `{m}` roles by re-checking the reasoning behind your selection and assign them to each agent to solve the problem.

---

## B.2 DYNAMIC AGENT

---

**Prompt for DYNAMIC AGENT**

You are a `{role}`. Your task is to solve a {coding/math/agentic} problem. Here is the problem that you have to solve:
`problem_dict[query['problem_id']]['description']`

---

**If external retrieval:** You were also given a couple of similar problems to the problem above along with their reasoning and solutions to aid you in solving the problem at hand. Here are the similar problems you were given:
`retrieved_dict[query['problem_id']]['retrieval_text']`

**If self-retrieval:** Further, recall a relevant and distinct problem (different from the problem mentioned above) along with its reasoning and solution.

---

And here was your original response:
`query[['role']]['original_thought','original_response']`

---

**If iteration $i \geq 1$ (i.e., $\mathcal{D}_S$ is not empty):**
Also here is the leading responses with execution results from the response store:
`response_dict['role','thought','response','score']`

---

**If coding task:**
Think carefully about where you went wrong, relating with responses in the response store. Then, try to fix the solution producing a thought later reply with a {Python} solution to be executed and judged again.

Make sure to wrap your code in ```python ``` block and Markdown delimiters, and include exactly one block of code with the entire solution (in the final code step).

---

**If math/agentic task:**
Think carefully about where you went wrong, relating with responses in the response store. Then, try to fix the solution producing a thought later reply with a solution to be executed and judged again. You can integrate required tools to execute the task while replying your solution if required.

Make sure to wrap your final answer in `\boxed{}` block with the entire solution (in the final answer step).

---

### B.3 JUDGE AGENT

---

**Prompt for JUDGE AGENT**

You are a judge. Your task is to judge the candidate solution of a {coding/math/agentic} problem. Here is the problem for which the candidate solution you have to judge:
`problem_dict[query['problem_id']]['description']`

---

**If coding task:**
And here is the candidate response along with test cases against which to judge:
`query[['candidate_role']]`
`['candidate_thought','candidate_response','test_case']`

Please produce a score (based on the number of test cases passed) with reasoning behind your judgement of the candidate solution to the problem.

---

**If math/agentic task:**
And here is the candidate response which to judge:
`query[['candidate_role']]['candidate_thought','candidate_response'`

Please produce a score (if the response is correct, it should be 1 otherwise should be 0) with reasoning behind your judgement of the candidate solution to the problem.

---

### B.4 VERIFIER AGENT

---

**Prompt for VERIFIER AGENT**

Your are an answer extractor. Your task is to extract answer from the response to a {coding/math/agentic} problem. Here is the response for which the answer you have to extract:
`response_dict[query{'role'}]{'thought', 'response', 'score'}`

---

**If coding task:**
Please extract the answer from inside ` ```python ``` ` block from the response.

---

**If math/agentic task:**
Please extract the answer from inside `\boxed{}` block from the response.

---

### B.5 REASONING SEGMENTATION

---

**Prompt for INITIAL REASONING SEGMENTATION**

Below is a chain of thought for solving a question. Figure out what are the different reasoning patterns that are used like problem rephrasing, new approach, self-evaluation, verification, backtracking, subgoal setup, solving a simpler problem first, and more. Then your task is to segment the entire chain of thought into different reasoning patterns. Rewrite the chain of thought in the following format:

```
<pattern> pattern name </pattern>
<content> the entire text
that corresponds to the pattern </content>
```
Thought: {thoughts}

---

---

**Prompt for FINAL REASONING SEGMENTATION**

Below is a thought for solving a question. For the segment between the `<unannotated>` and `</unannotated>` tags, figure out what is reasoning pattern used in that segment like problem rephrasing, new approach, self-evaluation, verification, backtracking, subgoal setup, solving a simpler problem first, or something else. Then your task is to identify the reasoning pattern used in the unannotated segment. Rewrite the unannotated segment in the following format:

```
<content> The text within the unannotated segment
that corresponds to the pattern. </content>
<reasoning> Reasoning for
what the pattern should be for the content </reasoning>
<pattern> *single* pattern name </pattern>
```

Thought: {thoughts}

---

## C  QUALITATIVE EXAMPLES

The subsequent section presents some qualitative examples–agentic interaction breakdown on how 𝕏olver can approach and solve a problem of math, agentic and coding tasks when it encounters. These examples illustrate how 𝕏olver initiates reasoning from external or self-retrieved exemplars, engages in multi-agent collaboration, and incrementally accumulates experiences through inter-agent propagation and refinement. The full interaction trace highlights 𝕏olver's ability to iteratively decompose, solve, and adapt solutions across reasoning steps, showcasing its capacity for dynamic knowledge construction and generalizable problem solving.

---

**Example for CORRECT MATH TASK** GREEN : CORRECT ANSWER. RED : WRONG ANSWER.

Query:  Bob is in charge of doing laundry for a large hotel. Each room has two sheets, one comforter, twice as many pillow cases as sheets and twice as many towels as pillow cases. How many pieces of laundry are there in 80 rooms? Make sure to wrap your final answer in `\boxed{}` block with the entire solution (in the final answer step).

**Iteration 1**

Mathematical Modeler:  Each room contains 2 sheets, 1 comforter, 2 pillow cases (which is twice the number of sheets), and 4 towels (which is twice the number of pillow cases). Hence, the total number of laundry items in each room is:
$2 + 1 + 2 + 4 = 9$ pieces of laundry.
When considering 80 rooms, the total number of pieces of laundry is:
$80 \times 9 = 720$ pieces of laundry.
Thus, the final answer is: $\boxed{720}$.

Judge:  The candidate's solution contains an error in calculating the number of pillow cases and towels per room. Here's the breakdown:
Sheets: 2 (correct). Comforter: 1 (correct). Pillow Cases: The problem states "twice as many pillow cases as sheets." Since there are 2 sheets, pillow cases should be $2 \times 2 = 4$. The candidate incorrectly used 2 pillow cases (half the required amount). Towels: The problem states "twice as many towels as pillow cases." With the correct pillow case count (4), towels should be $2 \times 4 = 8$. The candidate used 4 towels (based on their incorrect pillow case count of 2).
The correct per-room total is:
$2 \text{ (sheets)} + 1 \text{ (comforter)} + 4 \text{ (pillow cases)} + 8 \text{ (towels)} = 15$
items. The total for 80 rooms is:
$80 \times 15 = 1200$.

The candidate's answer is incorrect due to miscalculating pillow cases and towels.

---

Score: 0.

**Numerical Analyst:** For each room, Bob needs to wash 2 sheets, 1 comforter, 4 pillow cases (since $2 \times 2 = 4$), and 8 towels (since $4 \times 2 = 8$). Therefore, for each room, the total amount of laundry Bob needs to wash is:
$2 + 1 + 4 + 8 = 15$ pieces of laundry.
For 80 rooms, the total number of pieces of laundry Bob must wash is:
$80 \times 15 = 1200$ pieces of laundry.
Thus, the final answer is: $\boxed{1200}$.

**Judge:** The candidate correctly interprets the problem's relationships: sheets: 2 (given), comforter: 1 (given), pillow cases: twice the sheets, $2 \times 2 = 4$ (correct), towels: twice the pillow cases, $2 \times 4 = 8$ (correct). Total per room: $2 + 1 + 4 + 8 = 15$ (correct). Total for 80 rooms: $80 \times 15 = 1200$ (correct). All steps align with the problem's specifications, and the final answer is accurate. The candidate's solution is error-free.

Score: 1.

**Mathematician:** Every room is equipped with 2 sheets, 1 comforter, 4 pillow cases (calculated as $2 \times 2$), and 8 towels (since $4 \times 2 = 8$). Summing these, the total number of laundry items for one room is:
$2 + 4 + 8 = 14$ pieces of laundry.
Multiplying by the number of rooms, the total quantity for 80 rooms becomes:
$80 \times 14 = \boxed{1120}$.

**Judge:** The candidate incorrectly omitted the comforter in their final sum. Here's the breakdown: sheets: 2 (correct), comforter: 1 (given but not included in the candidate's calculation), pillow cases: $2 \times 2 = 4$ (correct), towels: $2 \times 4 = 8$ (correct). The candidate's total per room was $2 + 4 + 8 = 14$, but the correct total should include the comforter: $2 + 1 + 4 + 8 = 15$. Thus, the total for 80 rooms should be $80 \times 15 = 1200$, not 1120. The error stems from excluding the comforter in the summation.

Score: 0.

$\mathcal{D}_{S_1}$ rank: [Numerical Analyst, Mathematical Modeler, Mathematician]

**Iteration 2**

**Mathematical Modeler:** Each room contains 2 sheets, 1 comforter, 2 pillow cases (which is twice the number of sheets), and 4 towels (which is twice the number of pillow cases). Hence, the total number of laundry items in each room is:
$2 + 1 + 4 + 8 = 15$ pieces of laundry.
When considering 80 rooms, the total number of pieces of laundry is:
$80 \times 15 = 1200$ pieces of laundry.
Thus, the final answer is: $\boxed{1200}$.
This confirms the accuracy of the revised calculation.

**Judge:** The candidate's final answer is correct, and the mathematical calculations align with the problem's requirements: 1. Sheets: 2 (correct). 2. Comforter: 1 (included in the total). 3. Pillow Cases: The problem states "twice as many pillow cases as sheets." Since there are 2 sheets, this should be $2 \times 2 = 4$. The candidate correctly uses 4 pillow cases in their total calculation (despite a typographical inconsistency in the explanation, where they initially wrote "2 pillow cases"). 4. Towels: "Twice as many towels as pillow cases" gives $2 \times 4 = 8$ (correct).

The candidate's total per room is $2 + 1 + 4 + 8 = 15$ (correct), leading to $80 \times 15 = 1200$ (correct).

While the explanation contains a minor inconsistency in wording ("2 pillow cases" instead of 4), the numerical calculations are accurate. Since the final answer is correct and the mathematical steps resolve the inconsistency, the solution is valid.

Score: 1.

**Numerical Analyst:** Based on the information provided, we know that each room includes 2 sheets, 1 comforter, 4 pillow cases (computed as $2 \times 2$), and 8 towels (since $4 \times 2 = 8$). Adding these together, the total number of laundry items required for one room is:
$2 + 1 + 4 + 8 = 15$ pieces.
Multiplying by the total number of rooms, the overall amount of laundry for 80 rooms is:
$80 \times 15 = \boxed{1200}$.

Although the other agents correctly determined the number of laundry items required for each room, there was an error in the concluding step–calculate the total number of laundry pieces.

**Judge:** The candidate's calculations and final answer are correct: sheets: 2 (given), comforter: 1 (given), pillow cases: $2 \times 2 = 4$ (twice the sheets), towels: $2 \times 4 = 8$ (twice the pillow cases), total per room: $2 + 1 + 4 + 8 = 15$ (correct), total for 80 rooms: $80 \times 15 = 1200$ (correct).

Score: 1.

**Mathematician:** According to the provided details, every room is equipped with 2 sheets, 1 comforter, 4 pillow cases (calculated as $2 \times 2$), and 8 towels (since $4 \times 2 = 8$). Summing these, the total number of laundry items for one room is:
$1 + 4 + 8 = 13$ pieces of laundry.
Multiplying by the number of rooms, the total quantity for 80 rooms becomes:
$80 \times 13 = \boxed{1040}$.

**Judge:** The candidate incorrectly omitted the sheets in their final sum. Here's the breakdown: 1. Sheets: 2 (given but not included in the candidate's calculation). 2. Comforter: 1 (included). 3. Pillow Cases: $2 \times 2 = 4$ (correct). 4. Towels: $2 \times 4 = 8$ (correct). The candidate's total per room was $1 + 4 + 8 = 13$, but the correct total should include the sheets: $2 + 1 + 4 + 8 = 15$. Thus, the total for 80 rooms should be $80 \times 15 = 1200$, not 1040. The error stems from excluding the sheets in the summation.

Score: 0.

$\mathcal{D}_{S_2}$ rank: [Mathematical Modeler, Numerical Analyst, Mathematician]

**Iteration 3**

**Mathematical Modeler:** From the details provided, it is clear that each room contains 2 sheets, 1 comforter, 4 pillow cases (as $2 \times 2 = 4$), and 8 towels (calculated as $4 \times 2 = 8$). Adding these together, the total number of laundry items per room amounts to:
$2 + 1 + 4 + 8 = 15$ items.
Consequently, for 80 rooms, the total number of laundry pieces becomes:
$80 \times 15 = \boxed{1200}$.

I agree with the other agent's corrected calculation.

**Judge:** The candidate's solution is mathematically accurate and complete: sheets: 2 (correct), comforter: 1 (correct), pillow cases: $2 \times 2 = 4$ (twice the sheets), towels: $2 \times 4 = 8$ (twice the pillow cases), total per room: $2 + 1 + 4 + 8 = 15$ (correct), total for 80 rooms: $80 \times 15 = 1200$ (correct).

Score: 1.

**Numerical Analyst:** Given the information, we know that each room includes 2 sheets, 1 comforter, 4 pillow cases (computed as $2 \times 2$), and 8 towels (since $4 \times 2 = 8$). Adding these together, the total number of laundry items required for one room is:
$2 + 1 + 4 + 8 = 15$ pieces.
Multiplying by the total number of rooms, the overall amount of laundry for 80 rooms is:
$80 \times 15 = \boxed{1200}$.

Although the other agent correctly determined the number of laundry items required for each room, there was an error in the concluding step. Specifically, they incorrectly used $80 \times 13$ instead of the appropriate multiplication of $80 \times 15$ to calculate the total number of laundry pieces.

Judge: The candidate's solution is fully correct: sheets: 2 (given), comforter: 1 (given), pillow cases: $2 \times 2 = 4$ (twice the sheets), towels: $2 \times 4 = 8$ (twice the pillow cases), total per room: $2 + 1 + 4 + 8 = 15$ (correct), total for 80 rooms: $80 \times 15 = 1200$ (correct). All steps align with the problem's specifications, and the final answer is accurate. No errors were made.

Score: 1.

Mathematician: According to the given details, each room contains 2 sheets, 1 comforter, 4 pillow cases (calculated as $2 \times 2$), and 8 towels (determined by $4 \times 2$). Summing these amounts, the total number of laundry items per room is:
$2 + 1 + 4 + 8 = 15$ pieces of laundry.
To find the total for 80 rooms, we multiply:
$80 \times 15 = \boxed{1200}$.

I concur with the other agents' results and the computations they presented.

Judge: The candidate's solution is fully correct and aligns with the problem's requirements: sheets: 2 (given), comforter: 1 (given), pillow cases: $2 \times 2 = 4$ (twice the sheets), towels: $2 \times 4 = 8$ (twice the pillow cases), total per room: $2 + 1 + 4 + 8 = 15$ (correct), total for 80 rooms: $80 \times 15 = 1200$ (correct). All calculations are accurate, and the final answer matches the problem's constraints.

Score: 1.

$\mathcal{D}_{S_3}$ rank: [Mathematician, Mathematical Modeler, Numerical Analyst]

---

Example for CORRECT CODING TASK GREEN : CORRECT SOLUTION. RED : WRONG SOLUTION.

Query: Yarik is a big fan of many kinds of music. But Yarik loves not only listening to music but also writing it. He likes electronic music most of all, so he has created his own system of music notes, which, in his opinion, is best for it.

Since Yarik also likes informatics, in his system notes are denoted by integers of $2^k$, where $k \geq 1$ — a positive integer. But, as you know, you can't use just notes to write music, so Yarik uses combinations of two notes. The combination of two notes $(a, b)$, where $a = 2^k$ and $b = 2^l$, he denotes by the integer $a^b$.

For example, if $a = 8 = 2^3$, $b = 4 = 2^2$, then the combination $(a, b)$ is denoted by the integer $a^b = 8^4 = 4096$. Note that different combinations can have the same notation, e.g., the combination $(64, 2)$ is also denoted by the integer $4096 = 64^2$.

Yarik has already chosen $n$ notes that he wants to use in his new melody. However, since their integers can be very large, he has written them down as an array $a$ of length $n$, then the note $i$ is $b_i = 2^{a_i}$. The integers in array $a$ can be repeated.

The melody will consist of several combinations of two notes. Yarik was wondering how many pairs of notes $b_i, b_j$ $(i < j)$ exist such that the combination $(b_i, b_j)$ is equal to the combination $(b_j, b_i)$. In other words, he wants to count the number of pairs $(i, j)$ $(i < j)$ such that $b_i^{b_j} = b_j^{b_i}$. Help him find the number of such pairs.

Input

The first line of the input contains one integer $t$ $(1 \leq t \leq 10^4)$ — the number of test cases.

The first line of each test case contains one integer $n$ $(1 \leq n \leq 2 \cdot 10^5)$ — the length of the arrays.

The next line contains $n$ integers $a_1, a_2, \ldots, a_n$ $(1 \leq a_i \leq 10^9)$ — array $a$.

It is guaranteed that the sum of $n$ over all test cases does not exceed $2 \cdot 10^5$.

Output

For each test case, output the number of pairs that satisfy the given condition.

Sample Input 1:

5

1

2

4

3 1 3 2

2

1000 1000

3

1 1 1

19

2 4 1 6 2 8 5 4 2 10 5 10 8 7 4 3 2 6 10

Sample Output 1:

0
2
1
3
19

Make sure to wrap your code in ```python ``` block and Markdown delimiters, and include exactly one block of code with the entire solution (in the final code step).

**Iteration 1**

Implementation Specialist: To solve this problem, we need to determine the number of pairs of notes $(b_i, b_j)$ such that $b_i^{b_j} = b_j^{b_i}$. The notes are given as $b_i = 2^{a_i}$, where $a_i$ is an integer. The key insight is to recognize the conditions under which the equation $b_i^{b_j} = b_j^{b_i}$ holds. This equation can be broken down into two main cases. The first case is when the values are the same, meaning $a_i = a_j$. In this situation, the equation trivially holds because both sides will be identical. The number of such pairs is given by the combination formula $C(n, 2) = \frac{n(n-1)}{2}$ for each group of identical values. The second case occurs when the values are different but satisfy a special condition: the equation holds when one value is 1 and the other is 2. This is derived from the mathematical simplification of the equation $a_i \cdot 2^{a_j} = a_j \cdot 2^{a_i}$, which only holds true for $a_i = 1$ and $a_j = 2$ (or vice versa) when $a_i \neq a_j$.

```python
import sys
from collections import Counter

def main():
    # Read all input tokens at once
    x = sys.stdin.read().split()
    i = 0  # Index to track position in input list

    t = int(x[i])  # Number of test cases
    i += 1
```

```
        for _ in range(t):
            n = int(x[i])  # Number of elements in this test case
            i += 1

            # Extract the list of integers for current test case
            l = list(map(int, x[i:i+n]))
            i += n

            # Count frequencies of each element
            c = Counter(l)

            s = 0  # Sum of pairs

            # Sum pairs of identical elements: count choose 2
            for v in c.values():
                s += v * (v - 1) // 2

            # Add pairs formed by one element '1' and one element '2'
            s += c.get(1, 0) * c.get(2, 0)

            print(s)

if __name__ == "__main__":
    main()
```

Judge: There are 10 test cases. The candidate passes 5 test cases.

Score: 5.

Programmer: To solve this problem, we need to determine the number of pairs of notes such that the combination of two notes $(b_i, b_j)$ results in the same integer as the combination $(b_j, b_i)$. The notes are given as $b_i = 2^{a_i}$, and we are looking for pairs $(i, j)$ where $b_i^{b_j} = b_j^{b_i}$. The condition $b_i^{b_j} = b_j^{b_i}$ can be simplified to checking if $a_i \cdot 2^{a_j} = a_j \cdot 2^{a_i}$, which holds true in two scenarios: first, when $a_i = a_j$, producing identical pairs; and second, when one value is 1 and the other is 2, creating valid cross pairs. To count the identical pairs, for each unique value $x$ in the array, we use the combination formula $\binom{count}{2}$, where $count$ is the frequency of $x$. For cross pairs, which occur specifically between values 1 and 2, the number of such pairs is calculated as the product of the counts of 1 and 2.

```python
import sys
from collections import defaultdict

def main():
    x = sys.stdin.read().split()  # Read all input tokens
    i = 0  # Input index pointer

    t = int(x[i])  # Number of test cases
    i += 1

    for _ in range(t):
        n = int(x[i])  # Number of elements
        i += 1

        # List of integers for this test
        l = list(map(int, x[i:i+n]))
        case
        i += n

        f = defaultdict(int)  # Frequency dictionary

        # Count occurrences of each number
```

```
        for v in l:
            f[v] += 1

        s = 0  # Count pairs of identical elements

        # Sum pairs from frequencies (nC2)
        for v in f.values():
            s += v * (v - 1) // 2

        # Count pairs formed by one '1' and one '2'
        c = f.get(1, 0) * f.get(2, 0)

        print(s + c)

if __name__ == "__main__":
    main()
```

Judge: There are 10 test cases. The candidate passes 5 test cases.

Score: 5.

Data Structure Specialist: To solve this problem, we need to determine the number of pairs of music notes, represented by integers of the form $2^k$, such that the combination of two notes $(b_i, b_j)$ results in the same value as $(b_j, b_i)$. In other words, we seek pairs $(i, j)$ where $b_i^{b_j} = b_j^{b_i}$. By applying logarithms and using the properties of exponents, for notes defined as $b_i = 2^{a_i}$ and $b_j = 2^{a_j}$, the condition simplifies to $a_i \cdot 2^{a_j} = a_j \cdot 2^{a_i}$. Two key observations arise from this: first, if $a_i = a_j$, the equation holds true, meaning all pairs of identical values satisfy the condition; second, the equation also holds when one value is 1 and the other is 2, since $1 \cdot 2^2 = 2 \cdot 2^1$ simplifies to $4 = 4$. The algorithm to find the number of such pairs involves two parts. For pairs with the same value $v$, we count the number of pairs using the combination formula $C(n, 2) = \frac{n(n-1)}{2}$, where $n$ is the frequency of $v$. For cross-value pairs where one note corresponds to 1 and the other to 2, the number of valid pairs is simply the product of their counts.

```python
import sys
from collections import Counter

def main():
    x = sys.stdin.read().split()  # Read all input tokens at once
    i = 0  # Input index pointer

    t = int(x[i])  # Number of test cases
    i += 1

    for _ in range(t):
        n = int(x[i])  # Number of elements
        i += 1

        # List of integers for this test
        l = list(map(int, x[i:i+n]))
        case
        i += n

        c = Counter(l)  # Frequency counter

        c1 = c.get(1, 0)  # Count of 1's
        c2 = c.get(2, 0)  # Count of 2's

        x = 0

        # Combine counts of 1 and 2, calculate pairs among them
```

```
        m = c1 + c2
        x += m * (m - 1) // 2

        # Calculate pairs for other numbers except 1 and 2
        for k in c:
            if k not in (1, 2):
                v = c[k]
                x += v * (v - 1) // 2

        print(x)

if __name__ == "__main__":
    main()
```

Judge: There are 10 test cases. The candidate passes 5 test cases.

Score: 5.

$\mathcal{D}_{S_1}$ rank: [Implementation Specialist, Programmer, Data Structure Specialist]

**Iteration 2**

Implementation Specialist: : After reviewing the solutions provided by other agents, I agree that to solve this problem, we need to determine the number of pairs of notes $(b_i, b_j)$ such that $b_i^{b_j} = b_j^{b_i}$, where each note is given by $b_i = 2^{a_i}$. We are interested in counting pairs $(i, j)$ with $i < j$ that satisfy this condition. By applying properties of exponents and substituting $b_i = 2^{a_i}$ and $b_j = 2^{a_j}$, the condition simplifies to checking when $\frac{2^{a_i}}{a_i} = \frac{2^{a_j}}{a_j}$. There are two key observations: first, if $a_i = a_j$, the condition is automatically met; second, pairs where one value is 1 and the other is 2 also satisfy the condition, since $\frac{2^1}{1} = 2$ and $\frac{2^2}{2} = 2$. The algorithm involves counting the frequency of each $a_i$, then calculating the number of valid pairs for identical values using the combination formula $\binom{f}{2}$, where $f$ is the frequency of the value. Finally, to account for the special case of pairs between 1 and 2, we add the product of their counts to the total.

```python
import sys
from collections import Counter

def main():
    x = sys.stdin.read().split()  # Read all input tokens at once
    i = 0  # Index pointer

    t = int(x[i])  # Number of test cases
    i += 1

    for _ in range(t):
        n = int(x[i])  # Number of elements in this test case
        i += 1

        l = list(map(int, x[i:i+n]))  # List of elements
        i += n

        c = Counter(l)  # Frequency count

        s = 0  # Sum of pairs

        # Calculate pairs from identical elements: nC2 for each count
        for v in c.values():
            s += v * (v - 1) // 2

        # Add pairs formed by one '1' and one '2'
        s += c.get(1, 0) * c.get(2, 0)
```

```
            print(s)

    if __name__ == "__main__":
        main()
    ```
```

Judge: There are 10 test cases. The candidate passes 6 test cases.

Score: 6.

**Programmer:** I agree with the other agents that, to solve this problem, we need to determine the number of valid pairs of notes such that the combination of two notes $(b_i, b_j)$ results in the same integer as the combination $(b_j, b_i)$. This is equivalent to finding pairs $(i, j)$ where $b_i^{b_j} = b_j^{b_i}$. The key insight is that this equation holds under two conditions: first, when the exponents $a_i$ and $a_j$ are equal, the pairs $(b_i, b_j)$ will always satisfy the equation; second, when one exponent is 1 and the other is 2, the equation also holds. To solve the problem, the approach involves three steps. First, use a frequency counter to count the occurrences of each value in the array. Second, for each value $x$, compute the number of valid pairs among identical values using the combination formula $\binom{f}{2}$, where $f$ is the frequency of $x$. Third, calculate the number of specific pairs where one value is 1 and the other is 2 by multiplying their counts.

```python
import sys
from collections import Counter

def main():
    t = int(sys.stdin.readline())  # Number of test cases

    for _ in range(t):
        # Number of elements
        n = int(sys.stdin.readline().strip())
        # List of integers
        l = list(map(int, sys.stdin.readline().split()))

        c = Counter(l)  # Frequency counter

        s = 0  # Sum of pairs

        # Calculate pairs among identical elements:
        nC2 for each frequency
        for v in c.values():
            s += v * (v - 1) // 2

        # Add pairs formed by one '1' and one '2'
        s += c.get(1, 0) * c.get(2, 0)

        print(s)

if __name__ == "__main__":
    main()
```

Judge: There are 10 test cases. The candidate passes 5 test cases.

Score: 5.

**Data Structure Specialist:** To solve this problem, we need to determine the number of valid pairs of notes such that the combination of two notes $(b_i, b_j)$ results in the same integer as the combination $(b_j, b_i)$. This is equivalent to finding pairs $(i, j)$ where $b_i^{b_j} = b_j^{b_i}$. The key insight comes from the equation $b_i^{b_j} = b_j^{b_i}$, where $b_i = 2^{a_i}$ and $b_j = 2^{a_j}$. This equation simplifies to $a_i \cdot 2^{a_j} = a_j \cdot 2^{a_i}$. To solve this efficiently, we follow several steps. First, for each $a_i$, we perform prime factorization,

expressing it as $2^y \times x$, where $x$ is the odd part of $a_i$ and $y$ is the exponent of 2 in its factorization. Next, we generate a key for each $a_i$ in the form $(x, y - a_i)$. This key is designed such that if two numbers $a_i$ and $a_j$ share the same key, they satisfy the equation $a_i \cdot 2^{a_j} = a_j \cdot 2^{a_i}$. We then use a frequency dictionary to count the occurrences of each key. Finally, for each key, we compute the number of valid pairs using the combination formula $\binom{f}{2}$, where $f$ is the frequency of that key.

```python
import sys
from collections import defaultdict

def main():
    x = sys.stdin.read().split()  # Read all input tokens at once
    i = 0  # Input index pointer

    t = int(x[i])  # Number of test cases
    i += 1

    for _ in range(t):
        n = int(x[i])  # Number of elements in this test case
        i += 1

        l = list(map(int, x[i:i+n]))  # List of integers
        i += n

        f = defaultdict(int)  # Frequency dictionary

        for v in l:
            a = v
            b = 0
            # Count how many times 2 divides the number
            while a % 2 == 0:
                a //= 2
                b += 1
            # Key is tuple of (odd part, difference of exponents)
            k = (a, b - v)
            f[k] += 1

        s = 0  # Result accumulator

        # Count pairs for each unique key using nC2 formula
        for c in f.values():
            s += c * (c - 1) // 2

        print(s)

if __name__ == "__main__":
    main()
```

Judge: There are 10 test cases. The candidate passes all 10 test cases with correct result.

Score: 10.

$\mathcal{D}_{S_2}$ rank: [Data Structure Specialist, Implementation Specialist, Programmer]

### Iteration 3

Implementation Specialist: I concur with the other agent that to tackle this problem, our goal is to find the number of valid pairs of notes $(b_i, b_j)$ such that raising one note to the power of the other yields the same result when the order is reversed; that is, pairs $(i, j)$ satisfying $b_i^{b_j} = b_j^{b_i}$. Given that each note can be expressed as $b_i = 2^{a_i}$, the problem reduces to analyzing the equation $a_i \cdot 2^{a_j} = a_j \cdot 2^{a_i}$. To approach this effectively, we begin by decomposing each exponent $a_i$ into its prime factors, specifically

writing it as $2^y \times x$, where $x$ is the odd component and $y$ indicates the power of two within the factorization. Using this factorization, we create a unique key for each $a_i$ defined by the pair $(x, y - a_i)$. This construction ensures that any two exponents sharing the same key satisfy the original equation. After generating these keys for all elements, we count their frequencies using a dictionary. Finally, to determine the number of valid pairs, we apply the combination formula $\binom{f}{2}$ for each key, where $f$ is the frequency of that key, summing these results to obtain the total count of valid pairs.

```python
import sys
from collections import defaultdict

def main():
    d = sys.stdin.read().strip().split()  # Read and split all input
    t = int(d[0])  # Number of test cases
    p = 1  # Current position in input list

    for _ in range(t):
        n = int(d[p])  # Number of elements in this test case
        p += 1

        l = list(map(int, d[p:p+n]))  # Current list of elements
        p += n

        f = defaultdict(int)  # Frequency dictionary

        for v in l:
            o = v  # Original value
            c = 0  # Count of factors 2

            # Count how many times 2 divides the value
            while v % 2 == 0:
                v //= 2
                c += 1

            # Key: (odd part, count_twos - original_value)
            k = (v, c - o)
            f[k] += 1

        s = 0  # Result accumulator

        # Calculate pairs for each unique key using nC2
        for x in f.values():
            s += x * (x - 1) // 2

        print(s)

if __name__ == "__main__":
    main()
```

Judge: There are 10 test cases. The candidate passes all 10 test cases with correct result.

Score: 10.

Programmer: I agree with the other agent that to address this problem, we aim to find the number of valid pairs of notes $(b_i, b_j)$ such that the expression $b_i^{b_j}$ equals $b_j^{b_i}$. In other words, we seek pairs $(i, j)$ where the exponentiation relation $b_i^{b_j} = b_j^{b_i}$ holds true. Since each note is defined as $b_i = 2^{a_i}$, the equation simplifies to $a_i \cdot 2^{a_j} = a_j \cdot 2^{a_i}$. To efficiently solve this, we first factorize each exponent $a_i$ into the form $2^y \times x$, where $x$ represents the odd component and $y$ is the power of two in its prime factorization. Using this decomposition, we construct a key for each $a_i$ as $(x, y - a_i)$. This key is crafted so that if two exponents $a_i$ and $a_j$ share the same key, they satisfy the

equality condition. Next, we tally the frequency of each key using a dictionary or hash map. Finally, the number of valid pairs corresponding to each key is computed using the combination formula $\binom{f}{2}$, where $f$ denotes the frequency of that key. Summing these values yields the total count of valid pairs.

```python
import sys
from collections import defaultdict

def main():
    d = sys.stdin.read().strip().split()  # Read all input tokens
    t = int(d[0])  # Number of test cases
    p = 1  # Position pointer in input list

    for _ in range(t):
        n = int(d[p])  # Number of elements
        p += 1

        l = list(map(int, d[p:p+n]))  # List of elements
        p += n

        f = defaultdict(int)  # Frequency dictionary

        for v in l:
            b = v  # Base value
            c = 0  # Count of factors 2

            # Count how many times 2 divides b
            while b % 2 == 0:
                b //= 2
                c += 1

            # Key is tuple: (odd part, power difference)
            k = (b, c - v)
            f[k] += 1

        s = 0  # Result accumulator

        # Count pairs for each frequency using nC2 formula
        for x in f.values():
            s += x * (x - 1) // 2

        print(s)

if __name__ == "__main__":
    main()
```

Judge: There are 10 test cases. The candidate passes all 10 test cases with correct result.

Score: 10.

Data Structure Specialist: To solve this problem, the goal is to identify the number of valid pairs of notes $(b_i, b_j)$ for which the value of $b_i^{b_j}$ equals that of $b_j^{b_i}$. This translates to finding pairs $(i, j)$ satisfying the equation $b_i^{b_j} = b_j^{b_i}$. Given that each note $b_i$ can be expressed as $2^{a_i}$, the equation simplifies to $a_i \cdot 2^{a_j} = a_j \cdot 2^{a_i}$. To approach this efficiently, each exponent $a_i$ is factorized into the form $2^y \times x$, where $x$ denotes the odd part of $a_i$ and $y$ is the exponent of 2 in its prime factorization. Based on this factorization, we construct a unique key $(x, y - a_i)$ for each $a_i$. This key is carefully designed so that two numbers sharing the same key satisfy the given equality condition. Once the keys are generated, a frequency map is used to count how many times each key appears. Finally, for each key, the number of valid pairs is calculated using the combination formula $\binom{f}{2}$, where $f$ represents the frequency of that key. Summing these counts across all keys gives the total number of valid pairs.

```python
import sys
from collections import defaultdict

def main():
    d = sys.stdin.read().split()  # Read all input tokens
    i = 1  # Index pointer after reading t

    t = int(d[0])  # Number of test cases

    for _ in range(t):
        n = int(d[i])  # Number of elements
        i += 1

        l = list(map(int, d[i:i+n]))  # Current list of elements
        i += n

        f = defaultdict(int)  # Frequency dictionary

        for v in l:
            x = v  # Current value
            p = 0  # Count of factor 2

            # Count how many times 2 divides x
            while x % 2 == 0:
                x //= 2
                p += 1

            k = (x, p)  # Key as (odd part, power of two)
            f[k] += 1

        s = 0  # Result accumulator

        # Count pairs for each unique key using nC2 formula
        for c in f.values():
            s += c * (c - 1) // 2

        print(s)

if __name__ == "__main__":
    main()
```

While the previous codes are correct for the given problem constraints, it implicitly assumes that only pairs of equal values or $(1, 2)$ satisfy the equation later searching iteratively. However, explicitly grouping all valid pairs using a mathematical key, making it more robust and efficient for all scenarios, including possible edge cases not covered by the first approach.

Judge: There are 10 test cases. The candidate passes all 10 test cases with correct result.

Score: 10.

$\mathcal{D}_{S_3}$ rank: [Data Structure Specialist, Programmer, Implementation Specialist]

## D    EXTERNAL RETRIEVAL CORPUS FOR CODE

As shown in Figure 5, our coding external retrieval corpus has 156 fine grained algorithmic tutorials with relevant problems along with their solutions collections from 10 different categories (algebra, data structure, dynamic programming, string processing, linear algebra, combinatorics, numerical methods, geometry, graphs and miscellaneous).

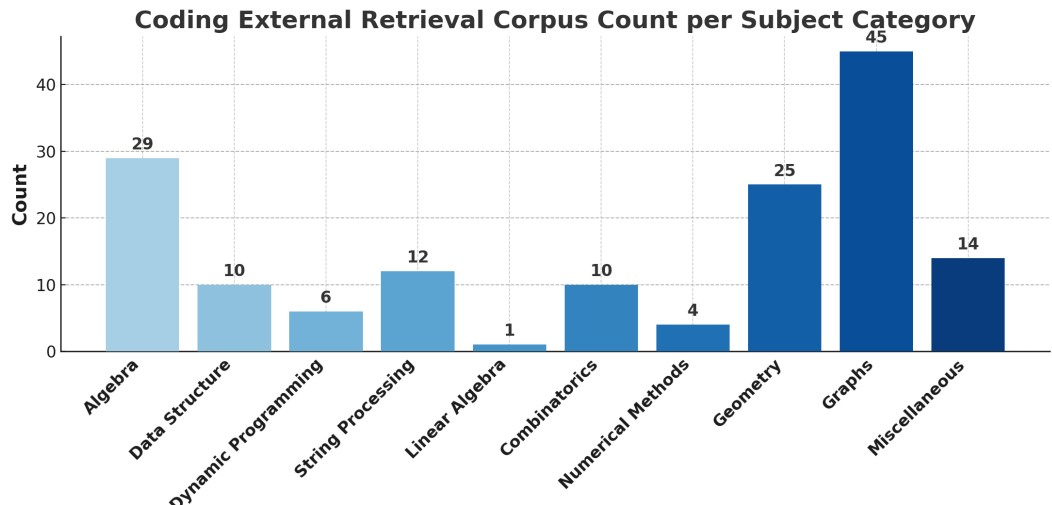

Figure 5: Coding External Retrieval Corpus Count per Subject Category.

## E    ADDITIONAL ANALYSIS

This section presents our multiple runs' results which we tested and list of roles which LLMs selected during the experiments.

### E.1    ADDITIONAL RESULTS

Table 2 presents results against some open-ended and multi-hop reasoning tasks, highlighting $\mathbb{X}$olver's effectiveness on diverse domains.

**Strong Gains Across Benchmarks.** Overall, $\mathbb{X}$olver consistently delivers significant improvements over the backbone LLMs' standard LongCoT prompting. Both the problem-specific $\mathbb{X}$olver (–) and the cross-problem $\mathbb{X}$olver (+) variants outperform their respective LongCoT baselines across all datasets. For example, with the o3-mini-medium backbone, $\mathbb{X}$olver (+) improves from 84.8 to 95.9 EM on SQuAD2.0, from 41.8 to 78.9 EM on 2WIKI, and from 46.6 to 77.0 EM on Bamboogle. Similarly, with the QWQ-32B backbone, $\mathbb{X}$olver (+) boosts EM from 79.4 to 90.2 on SQuAD2.0, from 34.4 to 70.2 on 2WIKI, and from 38.4 to 64.1 on Bamboogle.

**Surpassing Prior Agents.** Compared to prior frameworks such as Search-o1, $\mathbb{X}$olver demonstrates consistent and significant gains. With o3-mini-medium, $\mathbb{X}$olver (+) outperforms Search-o1 by +5.0 EM on SQuAD2.0 (95.9 vs. 90.9), +12.6 EM on 2WIKI (78.9 vs. 66.3), and +13.3 EM on Bamboogle (77.0 vs. 63.7). With QWQ-32B, $\mathbb{X}$olver (+) also yields clear improvements, surpassing Search-o1 by +5.2 EM on SQuAD2.0, +12.2 EM on 2WIKI, and +8.1 EM on Bamboogle.

**In Comparison to Leading LLMs.** Despite using weaker backbones, $\mathbb{X}$olver (+) matches or surpasses the strongest proprietary variants. For instance, with o3-mini-high, $\mathbb{X}$olver (+) achieves state-of-the-art results across all benchmarks: 97.5 EM on SQuAD2.0, 84.4 EM on 2WIKI, and 82.3 EM on Bamboogle—outperforming all previously reported results.

**Backbone Agnostic.** Improvements from $\mathbb{X}$olver are consistent across different backbone LLMs. Both o3-mini-medium and QWQ-32B benefit substantially from the framework, demonstrating its

model-agnostic design. On 2WIKI, for example, 𝕏olver (+) reaches 78.9 EM (o3-mini-medium) and 70.2 EM (QWQ-32B), a dramatic gain over the respective LongCoT baselines of 41.8 and 34.4.

**Effectiveness of Dynamic Episodic Memory.** While both variants excel, the cross-problem variant 𝕏olver (+) consistently outperforms the problem-specific version 𝕏olver (–) in all benchmarks. On average, episodic memory integration yields an improvement of +4.6 EM across datasets and backbones, with the largest gain being +6.3 EM on 2WIKI with o3-mini-medium (78.9 vs. 72.6).

**Scales with Backbone LLM's Strength.** 𝕏olver's performance scales consistently with the strength of its backbone. With o3-mini-high, 𝕏olver (+) achieves 97.5 EM / 98.8 F1 on SQuAD2.0, 84.4 EM / 97.6 F1 on 2WIKI, and 82.3 EM / 93.8 F1 on Bamboogle, setting new state-of-the-art results across all multihop retrieval benchmarks.

| Model | Appr. | SQuAD2.0 | | 2WIKI | | Bamboogle | |
|---|---|---|---|---|---|---|---|
| | | **EM** | **F1** | **EM** | **F1** | **EM** | **F1** |
| o3-mini-medium | LongCoT | 84.8 | 87.3 | 41.8 | 48.3 | 46.6 | 60.3 |
| | Search-o1 | 90.9 | 93.0 | 66.3 | 78.0 | 63.7 | 74.9 |
| | 𝕏olver(–) | 92.2 | 96.3 | 72.6 | 87.5 | 70.9 | 81.1 |
| | 𝕏olver (+) | **95.9** | **98.3** | **78.9** | **96.3** | **77.0** | **88.7** |
| QWQ-32B | LongCoT | 79.4 | 83.9 | 34.4 | 40.9 | 38.4 | 53.7 |
| | Search-o1 | 85.0 | 88.4 | 58.0 | 71.4 | 56.0 | 67.8 |
| | 𝕏olver(–) | 88.4 | 91.5 | 63.2 | 78.3 | 60.8 | 72.6 |
| | 𝕏olver(+) | 90.2 | 94.6 | 70.2 | 86.6 | 64.1 | 79.5 |
| o3-mini-high | 𝕏olver (+) | **97.5** | **98.8** | **84.4** | **97.6** | **82.3** | **93.8** |

Table 2: Comparison of 𝕏**olver** against SoTA reasoning agents across multihop retrieval tasks. Best results are boldfaced and second-best results are underlined. LongCoT: standard prompting for reasoning models.

## E.2 PERFORMANCE VARIANCE STATISTICS.

In this experiment on the variance of 𝕏olver performance, we tested 𝕏olver against multiple runs (16 for AIME '24 and 32 for AIME '25 and LiveCodeBench) in AIME and LiveCodeBench dataset. Results shows in Table 3 that it has small scale performance change with multiple runs which is a strong sign on the robustness of 𝕏olver.

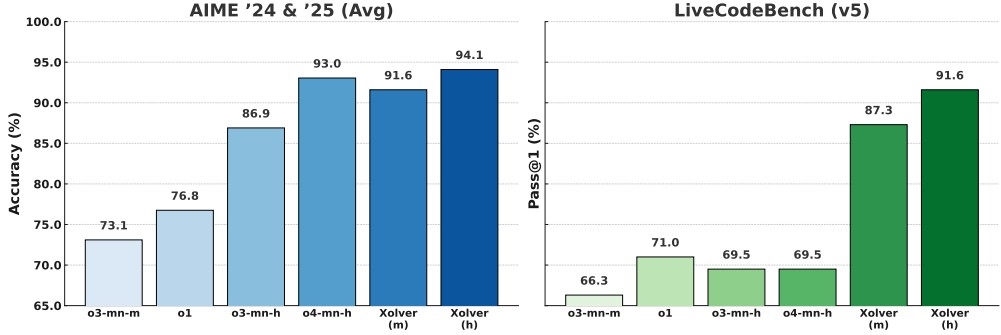

Figure 6: **Results Summary on AIME '24 (16 runs), AIME '25 and LiveCodeBench (32 runs).** Our framework 𝕏**olver**, built on `o3-mini-medium` and `o3-mini-high` backbones (denoted (m) and (h)), achieves up to 30.9% gain over the baseline and often outperforms leading models on both tasks.

As in Figure 6, 𝕏olver (m) achieves 91.6% average accuracy on the AIME '24 and '25 benchmarks—an 18.5-point gain over o3-mini-medium—while 𝕏olver (h) reaches 94.1%, outperforming o3-mini-high by 7.2 points. On LiveCodeBench, 𝕏olver (m) improves upon its base by 21 points (66.3% to 87.3%), with 𝕏olver (h) achieving 91.6%, a 22.1-point lift over o3-mini-high.

| Model | Appr. | AIME '24 | AIME '25 | LiveCodeBench (v5) |
|-------|-------|----------|----------|---------------------|
| o3-mini-medium | Xolver (-) | $87.2 \pm 1.2$ | $85.1 \pm 1.3$ | $79.6 \pm 1.0$ |
|  | Xolver (+) | $93.8 \pm 0.3$ | $89.4 \pm 0.7$ | $87.3 \pm 0.4$ |
| QWQ-32B | Xolver (-) | $89.9 \pm 0.8$ | $79.5 \pm 1.1$ | $76.2 \pm 0.9$ |
|  | Xolver (+) | $93.6 \pm 0.2$ | $82.7 \pm 0.8$ | $79.2 \pm 0.5$ |
| o3-mini-high | Xolver (+) | $94.4 \pm 0.6$ | $93.7 \pm 0.5$ | $91.6 \pm 0.3$ |

Table 3: Xolver average performance with multiple trials.

### E.3 IMPACT OF AGENT AND TOOL COUNT WITH ITERATIONS, AND EMERGING BENEFITS OF COLLABORATION

We analyze the effect of varying the number of agents , tools and reasoning iterations on Xolver's performance. In a controlled setup, we fix one variable (e.g., 3 agents, 4 tools or 2 iterations) and incrementally increase the other. As shown in Figure 3, performance improves consistently on both AIME '25 and LIVECODEBENCH with more agents or iterations, whereas Figure 7 (see Section 4) shows steadily improving performance

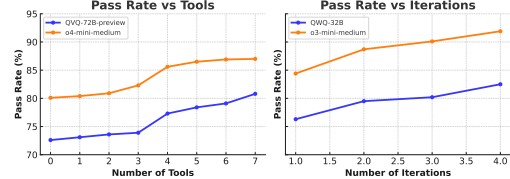

Figure 7: Impact of tools and iterations in Xolver on GAIA (left) and BAMBOOGLE (right).

on both GAIA and BAMBOOGLE with more tools or iterations, highlighting the advantage of collaborative and iterative problem solving.

To probe deeper, we conduct a budget-controlled experiment on the AIME '25 dataset, where the total reasoning budget (i.e., number of agents × number of iterations) is fixed. While iterative reasoning remains a crucial factor for Xolver's performance, we find that increasing the number of agents—particularly beyond a minimum of three—yields additional, emergent improvements, leading to over a 4% performance gain. This suggests that agent diversity and parallelism complement iterative depth, together producing stronger collaborative problem-solving benefits than either alone.

### E.4 EFFECT OF RETRIEVAL STRATEGIES ON XOLVER PERFORMANCE

We evaluate the impact of different retrieval strategies on Xolver by comparing three settings: (1) *External Retrieval*, where the model retrieves the top-$k$ (e.g., $k = 5$) most similar problems and their solutions from an external corpus using a BM25 retriever; (2) *Self-Retrieval*, where the model recalls the top-$k$ most similar problems and solutions from its own internal memory; and (3) *No Retrieval*, where neither external nor self-retrieval is used. As shown in Figure 8 (see Section 4), performance on both AIME '25 and LIVECODEBENCH follows the trend: *External Retrieval > Self-Retrieval > No Retrieval*, indicating that external retrieval significantly enhances Xolver's performance.

We note that for code tasks, although the external retrieval corpus contains solutions written in C++—a different language from the target Python—external retrieval still provides a substantial performance boost. Nonetheless, while self-retrieval results in a notable performance drop compared to external retrieval, it still outperforms the no-retrieval baseline with notable margins, serving as a viable alternative when external resources are unavailable.

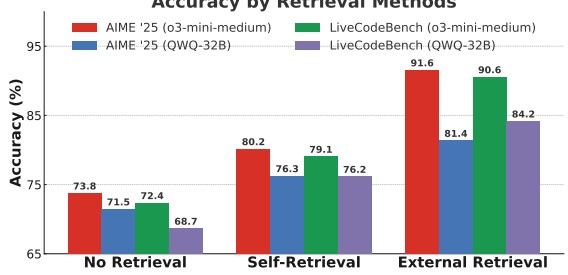

Figure 8: Impact of different retrievals in Xolver.

### E.5 FINE-GRAINED PERFORMANCE ANALYSIS

Figures shows that, on MATH-500 (Figure 9), LIVECODEBENCH (Figure 10) and GPQA DIAMOND (Figure 11), Xolver (both o3-mini-medium and QWQ-32B) consistently outperforms CHEATSHEET across nearly all seven subject categories, despite the latter relying on costly per-problem memory updates. The only exception is in *Number Theory*, where `o3-mini-medium` scores 99.2 compared to CHEATSHEET's 99.5. As for QWQ-32B,

Xolver achieves substantial accuracy gains over CheatSheet across all categories, with improvements of +9.0% in Prealgebra, +8.5% in Algebra, +11.0% in Number Theory, +8.5% in Counting and Probability, +8.8% in Geometry, +10.0% in Intermediate Algebra, and +7.5% in Precalculus. These consistent

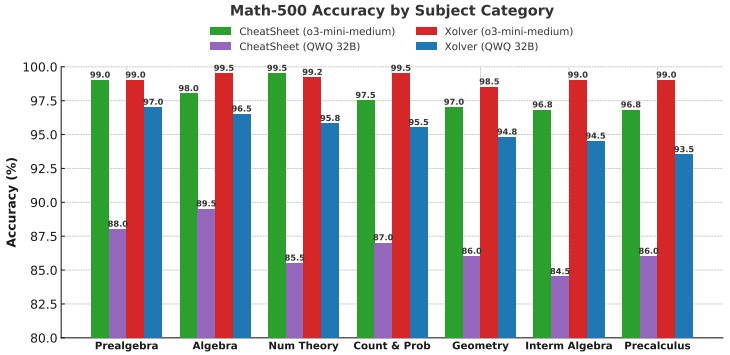

Figure 9: Fine-grained performance comparison in MATH-500.

gains highlight Xolver's strong performance across both symbolic and numerical reasoning. On LiveCodeBench, Xolver demonstrates even more pronounced gains. The o3-mini-medium variant achieves 95.6%, 90.4%, and 85.8% accuracy on Easy, Medium, and Hard problems respectively, significantly outperforming CodeSim by +4.5%, +11.9%, and a striking +32.3% margin on hard examples. Even with a weaker QWQ-32B backbone, Xolver (95.2%, 87.5%, 70.0%) surpasses all baselines and achieves similar gains. In contrast to CheatSheet and CodeSim, Xolver leverages multi-agent collaborations and holistic experience learning.

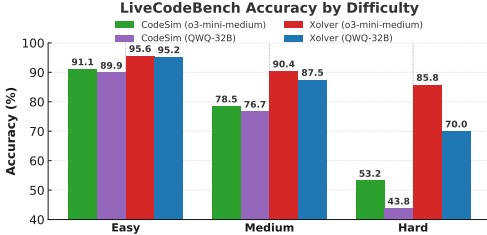

Figure 10: Performance comparison per difficulty levels in LiveCodeBench.

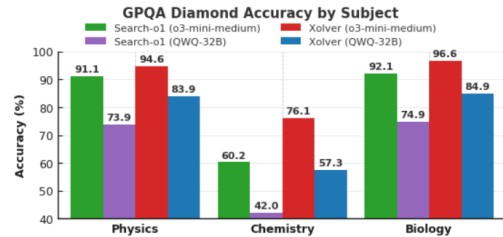

Figure 11: Fine-grained performance comparison in GPQA Diamond.

On GPQA Diamond, Xolver improves accuracy across all three subjects for both backbones (see Figure 11). With the o3-mini-medium backbone, Xolver attains 94.6% (Physics), 76.1% (Chemistry), and 96.6% (Biology), outperforming Search-o1 by +3.5%, +15.9%, and +4.5% points, respectively. Even with the weaker QVQ-32B backbone, Xolver reaches 83.9%, 57.3%, and 84.9%—gains of +10.0%, +15.3%, and +10.0% points over the corresponding Search-o1 results. These consistent, backbone-agnostic improvements highlight Xolver's strong performance on multi-disciplinary scientific QA task.

### E.6 CAN A SELF-JUDGE REPLACE A JUDGE AGENT?

We analyze the effect of different judging mechanisms on Xolver's performance by comparing two setups: (1) *self-judging*, where each dynamic agent evaluates its own response through self-reflection without altering its role, and (2) *external judging*, where a separate judge agent is used to assess the responses. We find that self-judging agents tend to be biased in favor of their own outputs, occasionally validating incorrect solutions. This self-bias leads to a noticeable drop in overall performance—specifically, a 7.4% decrease in coding tasks, 4% decrease in agentic tasks and a 8.7% decrease in math tasks, on average.

### E.7 COST ANALYSIS AND HOW LONG DO XOLVER AGENTS THINK?

In Figure 12, we report input, reasoning, and output token statistics for Xolver across all datasets. Token complexity scales as $O(m\mathcal{I})$ (with $m$ agents and $\mathcal{I}$ iterations), while runtime remains $O(\mathcal{I})$ because agents run in parallel. This is more efficient than self-consistency (Wang et al., 2022), which typically needs 32–64 generations per example, and than the CheatSheet baseline with $O(n^2)$ memory updates from usefulness estimation over prior examples. As a multi-agent system, Xolver spends most tokens on context sharing and inter-agent communication; about 25% go to core reasoning.

We also compare total token usage to the single-agent Search-o1, counting tokens with `tiktoken` (o3-mini-medium) and `AutoTokenizer` (QWQ-32B). Xolver uses roughly 1.5× more tokens, reflecting collaborative, iterative reasoning, yet this moderate increase yields substantial gains: +32.3% absolute on hard coding with o3-mini-medium and +9.05% accuracy across all Math-500 categories with QWQ-32B (see

Figure 12: Average numbers of token usage across datasets in Xolver.

Figure 9; Figure 10). Thus, Xolver 's slightly higher reasoning cost is well justified by its broad, superior performance.

### E.8   Impact of Data-Shuffling in Xolver (+) Performance

Xolver (+) updates its external memory incrementally after solving each new problem. To examine whether the order of test instances impacts performance, we conduct an ablation study by randomly shuffling the sequence of problems in each task. This helps determine if there is any dependency on the data order. During this experiment on the impact of shuffling data on Xolver performance, we randomly shuffled the test instances and conducted the experiment with 5 runs. Results shows in Table 4 that Xolver has limited performance change (STD $\sim$ 1) with shuffling data–a strong sign on the robustness of the framework.

| Mean $\pm$ STD | (With only 5 Runs, all STD $\sim$ 1) | | | | |
|---|---|---|---|---|---|
| Model | GSM8K | AIME '24 | AIME '25 | MATH-500 | LiveCodeBench (v5) |
| o3-mini-medium | $97.6 \pm 1.3$ | $92.2 \pm 0.4$ | $91.0 \pm 0.3$ | $98.3 \pm 0.6$ | $90.9 \pm 1.1$ |
| QWQ-32B | $97.2 \pm 0.6$ | $93.7 \pm 0.5$ | $82.7 \pm 2.0$ | $95.1 \pm 0.6$ | $83.6 \pm 1.6$ |

Table 4: Impact of using intermediate shared memory with shuffle of order in test set in X**olver**.

### E.9   More Error Analysis

In Figure 13 and Figure 14, we present an error analysis across math, code and agentic tasks that goes beyond simple accuracy or pass@1 metrics. While Xolver significantly improves reasoning and generation capabilities in these domains, both (o3-mini-medium and QWQ-32B equivalent) backbone LLMs can still produce solutions that are syntactically correct yet semantically flawed, resulting in failed executions due to incorrect reasoning, incomplete logic, unoptimized implementations, or misaligned tool usage. In

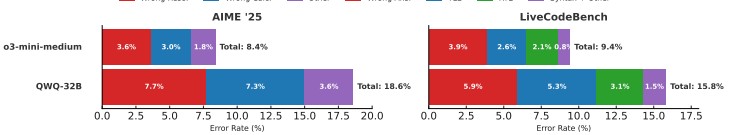

Figure 13: Xolver Math and Code error distribution.

agentic tasks, failure mode include mouse click, instructual, visual and other errors, whereas, code tasks include incorrect final code, time limit exceeded (TLE), runtime errors (RTE), and syntax issues. In math tasks, remaining errors are primarily due to flawed logical derivations or faulty intermediate calculations. Although Python-based tools are available, such calculation errors often occur when agents choose not to invoke these tools—highlighting that tool usage remains decoupled from the model's core reasoning process (see Appendix B for our prompt design). These findings provide insights for future improvements by exposing the variety of failure modes across domains, and further emphasize the importance of robust self-verification and refinement mechanisms, as employed by Xolver.

### E.10   Patterns in Xolver Reasoning Traces

To understand how 𝕏olver adapts its reasoning process to perform complex reasoning, we analyze the dynamics of reasoning pattern frequencies across difficulty

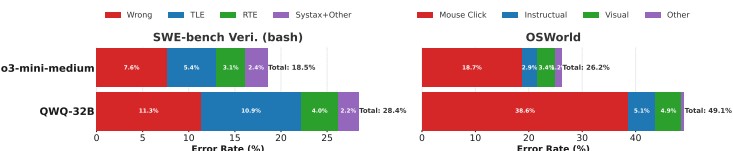

Figure 14: 𝕏olver Code and Agentic task error distribution.

levels in LIVECODEBENCH, as shown in Table 5. Our analysis reveals that 𝕏olver dynamically increases *self-evaluation* and *exploratory strategies* (e.g., trying new approaches) as problem difficulty grows. Correct solutions demonstrate a declining need for problem rephrasing and subgoal decomposition, indicating more direct and confident reasoning. In contrast, incorrect solutions show increased subgoal setup and rephrasing attempts—suggesting that the system recognizes failure and attempts recovery through restructuring. Compared to OPENCODEREASONING, which shows stagnation or regression in key patterns (e.g., self-evaluation), 𝕏olver exhibits robust and adaptive reasoning behavior, supported by multi-agent collaboration and judge feedback. This behavior highlights the generality and flexibility of 𝕏olver 's reasoning model.

| Reasoning Pattern | Correct Solutions | | Incorrect Solutions | |
|---|---|---|---|---|
| | Easy → Medium | Medium → High | Easy → Medium | Medium → High |
| **(a) OpenCodeReasoning** | | | | |
| Self-Evaluation (↓) | 0.39 → 0.37 | 0.37 → 0.34 | 0.36 → 0.37 | 0.34 → 0.31 |
| New Approach (↑) | 0.16 → 0.20 | 0.20 → 0.23 | 0.16 → 0.22 | 0.22 → 0.25 |
| Problem Rephrasing (↓↑) | 0.21 → 0.20 | 0.20 → 0.20 | 0.21 → 0.22 | 0.22 → 0.23 |
| Subgoal Setup (↓) | 0.13 → 0.12 | 0.12 → 0.10 | 0.13 → 0.10 | 0.10 → 0.10 |
| **(b) 𝕏olver** | | | | |
| Self-Evaluation (↑) | 0.35 → 0.38 | 0.38 → 0.40 | 0.35 → 0.37 | 0.32 → 0.35 |
| New Approach (↑) | 0.18 → 0.21 | 0.21 → 0.24 | 0.17 → 0.24 | 0.24 → 0.26 |
| Problem Rephrasing (↓↑) | 0.20 → 0.17 | 0.18 → 0.18 | 0.23 → 0.24 | 0.24 → 0.25 |
| Subgoal Setup (↓↑) | 0.14 → 0.13 | 0.13 → 0.11 | 0.11 → 0.12 | 0.11 → 0.11 |

Table 5: Demonstrating how the frequency of major reasoning pattern changes as problem difficulty increases. Green indicates statistically significant increases and red indicates significant decreases ($p < 0.05$). Gray boxes highlight opposing trends between OpenCodeReasoning (decrease) and 𝕏olver (increase). Direction arrows indicate the expected trend direction: ↑ = increase, ↓ = decrease, ↓↑ = mixed trend (minor decrease then elevated recovery), ↕ = fluctuating trend (major decrease then recovery). While solving problems, OpenCodeReasoning struggles at Self-Evaluation and Subgoal Setup whereas 𝕏olver overcomes it with increasing Self-Evaluation in both correct and incorrect solutions and elevated recovery in Subgoal Setup in incorrect solutions. Both OpenCodeReasoning and 𝕏olver adapts New Approach while struggles at Problem Rephrasing.

We employ the QWQ-32B model for the initial segmentation of the chain of thought, guided by the first prompt, subsequently, for segments that remain unannotated, a second segmentation pass is performed using the second prompt depicted in Section B.5. Reasoning patterns are identified by extracting content enclosed within the <pattern> tags. Verification and reasoning patterns are combined into a unified self-evaluation category. If the model assigns multiple patterns to a single segment, that segment is excluded due to ambiguity in pattern classification. For each generated output, we calculate the proportion of occurrences of each pattern relative to the total patterns present, resulting in a frequency vector representing pattern distribution per generation. In examining the relationship between pattern usage and problem difficulty, we compute the mean frequencies separately for correct and incorrect generations and assess significance through a t-test. To evaluate pattern prevalence on a per-problem basis, a binary matrix is constructed where rows correspond to problems and entries indicate whether a pattern is more common in correct (1) or incorrect (0) solutions. The statistical significance of these findings is evaluated using a binomial test.

### E.11 CASE-STUDY: HOW 𝕏OLVER ENHANCES REASONING

To further understand the reasoning and problem-solving strategies behind our multi-agent, iterative framework 𝕏olver, we conduct an in-depth analysis combining qualitative runtime inspection with controlled experiments. We begin by manually studying 𝕏**olver**'s agent interaction traces on AIME '25 and LiveCodeBench. These case studies reveal that at each iteration, dynamic

agents attempt to improve upon earlier failures by leveraging Judge agent feedback and by aligning with top-ranked outputs stored in the shared memory $\mathcal{D}_S$. This process results in progressively refined outputs, increased agent alignment, and eventual convergence toward correct solutions.

To verify this behavior systematically, we conduct a controlled experiment across both math and code tasks. We instantiate two dynamic agents with complementary strengths: a Coder agent and a Mathematician agent, each proficient in one domain but sub-optimal in the other. We then measure their performance and agreement across iterations—defined as the percentage of problems in which both

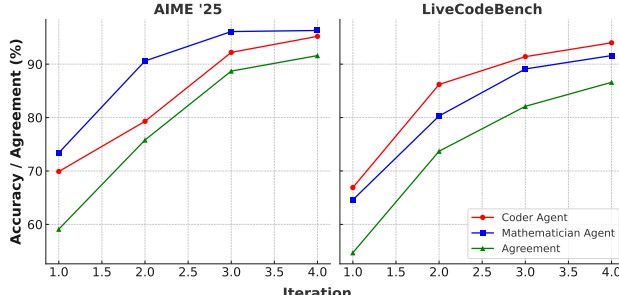

Figure 15: Agents Accuracy and Agreement over iterations.

agents independently produce the same correct answer (for math) or code that passes the same test cases (for code). As shown in Figure 15, both agents demonstrate consistent accuracy improvements over time, accompanied by a rising agreement rate. This not only illustrates mutual influence and learning-by-alignment but also validates the emergence of collaborative synergy.

Crucially, we observe that the presence of the Judge agent plays a vital role in this convergence process. When the Judge agent is removed—as shown in our first ablation—performance degrades significantly. These findings collectively affirm that $\mathbb{X}$**olver**'s iterative memory-sharing, feedback-driven refinement, and role-specialized agents contribute to its strong reasoning performance across domains, making it a compelling framework for general-purpose, self-improving problem solving.

### E.12 LIST OF ROLES OF SELECTED BY DYNAMIC AGENTS

Table 6 shows some selected specialized roles by the dynamic agents while testing on math, coding and agentic tasks along with their most frequently selected roles.

| (a) Specialized Roles | | |
|---|---|---|
| **Math** | **Coding** | **Agentic** |
| Problem Analyzer | Problem Analyzer | Snippet Extractor |
| Mathematical Modeler | Algorithm Designer | Web Search Strategist |
| Algorithm Designer | Solution Architect | Evidence Collector |
| Numerical Analyst | Implementation Specialist | Source Credibility Assessor |
| Symbolic Solver | Data Structure Specialist | Multi-Document Synthesizer |
| Mathematician | Optimization Engineer | Verification Critic |
| Computational Tools Specialist | Unit Tester | Browser Orchestrator |
| | Debugging Expert | Automation Specialist |
| | Programmer | Workspace Manager |
| | Debugging Expert | User Interface Navigator |
| | Code Reviewer | Rubric Aligner |
| | | Query Rewriter |
| **(b) Most Frequent Roles** | | |
| **Math** | **Coding** | **Agentic** |
| Mathematical Modeler | Algorithm Designer | Web Search Strategist |
| Numerical Analyst | Implementation Specialist | Query Rewriting Strategist |
| Symbolic Solver | Data Structure Specialist | Evidence Extractor |
| Mathematician | Programmer | Consistency Verifier |
| Computational Tools Specialist | Optimization Engineer | Multi-Document Synthesizer |

Table 6: List of math, coding and agentic roles selected by LLMs.

