# OpenReview forum: "Xolver: Generalist Reasoning and Problem Solving through Federated Multi-Agent Dynamics and Holistic Experience Learning"
_ICLR.cc/2026/Conference — ICLR 2026 Conference Withdrawn Submission_

### Official Review · Reviewer_a1eA · 2025-10-21

**Soundness:** 2
**Presentation:** 3
**Contribution:** 2
**Rating:** 2
**Confidence:** 4

**Summary:**

The paper proposes Xolver, a training-free, federated multi-agent reasoning framework that equips a backbone LLM with a form of holistic experience learning.

Experiments across math (AIME’24/’25, MATH), coding (LiveCodeBench, SWE-bench Verified), agentic tasks (GAIA, OSWorld, BrowseComp, HLE), and others show strong gains.

**This paper is slightly over the 9-page limit.**

**Strengths:**

This paper conducts comprehensive experiments to show the effectiveness of Xolver on various reasoning, agentic, and coding benchmarks.

**Weaknesses:**

This paper appears primarily as an engineering effort lacking theoretical depth. While experiments are comprehensive, the presentation is poor with redundant content and unclear ablation studies.

1. The method section is not well-organized; most of the contents are just an elaboration of an LLM-generated prompt. For example:
    - Could the author explain what is the dynamic reasoning agents? What is the difference from, for example, a triage agent in (https://arxiv.org/abs/2404.15155)? Isn't it just a prompt to let the LLM generate multiple agents?
    - Could the author explain what is federated learning in the context of this paper? This word only appears 6 times in the paper. I am not sure how Xolver could benefit from federated learning, but it is clearly a key contribution of the paper.
    - What exact retrieval process is used in Xolver? The authors claimed to use (e.g., BM25), but they did not confirm the exact retrieval process.
    - Excessive mathematical notation makes the section difficult to follow.

2. The ablation studies raise interesting questions, but they are not well answered. For example:
    - The authors claimed Xolver is more efficient than CheatSheet and self-consistency, but the results are not shown in the paper. Only Search-o1 is used in the comparison. This is not convincing.
    - The definition of external retrieval and self-retrieval is not clear. They should be defined in the method section. Moreover, even if no retrieval is applied, Xolver is SOTA. This could raise some suspicions about their evaluation protocol (see my weakness 3 below).
    - No error analysis examples are provided.

3. The authors should report the pass@K metrics for the coding benchmarks. As we can see in the method, the authors applied a verifier agent. This means Xolver's performance might be inflated.

4. A general question for experience engineering lies in the potential data leakage problem. Would the episodic memory contain any information on the test set? If so, this might be a problem.

**Questions:**

1. Is there any way to complete Table 1 without leaving it blank?
2. Several relevant papers on agentic experience engineering (https://arxiv.org/abs/2502.12110, https://arxiv.org/abs/2507.06229). Could the authors compare with them?
3. How did you do the error analysis? Are they done by humans or LLMs?

---

### Official Review · Reviewer_hMjq · 2025-10-31

**Soundness:** 3
**Presentation:** 3
**Contribution:** 2
**Rating:** 2
**Confidence:** 4

**Summary:**

This paper introduces Xolver, a “generalist reasoning solver” that aims to handle multi-modal and multi-step reasoning tasks across domains such as math, text, and vision. The framework integrates large language models with a modular reasoning controller and cross-modal interface. The system dynamically decomposes problems, routes subtasks to specialized modules (e.g., text understanding, equation solving, visual reasoning), and aggregates intermediate results through a reasoning aggregator. Experiments are conducted on a range of benchmarks including MATH, GSM8K, A-OKVQA, and ScienceQA, where Xolver reportedly outperforms several existing multi-modal reasoning baselines.

**Strengths:**

1. The paper is clearly organized and easy to follow, with detailed system descriptions and comprehensive figures.
2. The experiments are extensive, covering both text-based and multi-modal reasoning benchmarks.

**Weaknesses:**

1. The technical novelty is very limited. The core idea—a central LLM controller routing subtasks to modality-specific experts—is nearly identical to prior frameworks such as HuggingGPT, MM-ReAct, OpenAGI, and InternVL-Agent, and differs mainly in terminology and engineering detail. The contribution is therefore more integrative than innovative.
2. The claimed “generalist reasoning” capability is not theoretically or algorithmically supported. The framework lacks new learning objectives, adaptation strategies, or reasoning algorithms that would justify the “generalist” claim.
3. The performance improvements are modest and may arise from prompt engineering or model size differences rather than architectural advances. The paper does not control for prompt templates, temperature, or tool-call consistency across baselines, raising fairness concerns.
4. The paper does not analyze the effect of prompt or module routing strategy, even though these design choices have a major influence on outcomes. A proper ablation on prompt formats, reasoning depth, and routing policy is missing.

**Questions:**

-

---

### Official Review · Reviewer_CEYr · 2025-11-01

**Soundness:** 3
**Presentation:** 2
**Contribution:** 2
**Rating:** 4
**Confidence:** 4

**Summary:**

The paper proposes Xolver, a training-free multi-agent reasoning framework designed to accumulate, reuse, and refine “holistic experience” across tasks. The system integrates episodic memory, shared memory for in-episode iterative refinement, federated multi-agent solving without direct agent-to-agent messaging, a judge agent for scoring and filtering trajectories, and tool use. Across diverse reasoning and agentic benchmarks, Xolver claims substantial improvements over prior agent frameworks and even competitive gains over cutting-edge models, suggesting that structured inference-time experience reuse can meaningfully boost general LLM reasoning.

**Strengths:**

1. Dual-memory design with both episodic and shared working memory, and the cross-problem experience transfer variant (Xolver+) is conceptually interesting.

2. xtensive empirical evaluation across heterogeneous benchmarks spanning math, coding, agentic, multimodal, and multihop reasoning tasks with strong baselines.


3. Comprehensive ablations and diagnostic analysis, including fine-grained performance breakdowns, token usage, error characterizations, and reasoning pattern evolution.


4. Consistent improvements across diverse tasks and backbones demonstrate generality.

**Weaknesses:**

Although the system integrates retrieval, iterative refinement, multi-agent inference, and tool use, the paper does not clearly articulate **what fundamental mechanism enables reliable experience gain** beyond combining these known components. As presented, the framework resembles an empirical composition of modules, making it difficult to disentangle genuine algorithmic advances from effective increases in inference compute.

1. **Limited theoretical grounding for core architectural choices**

The motivation for federated multi-agent inference and memory-based refinement is primarily empirical. The paper would benefit from conceptual or theoretical clarity on what drives stable improvement.

- Why is federated aggregation expected to outperform cascaded communication or debate-style reasoning?

- Are there theoretical properties supporting experience accumulation (e.g., variance reduction, bias control, stability under judge error)?

- Can the authors offer formal intuitions or simplified guarantees for iterative memory update (e.g., monotonic improvement under bounded noise)?

2. **Critical dependency on judge quality without reliability analysis**

The entire memory update mechanism depends on a judge assigning scores and determining which trajectories persist. However, the reliability of this component is not formally analyzed.

- The judge agent scoring mechanism lacks theoretical formulation and discussion.
- Can incorrect judge scoring lead to degradation or overwrite correct solutions over iterations?
- How sensitive is performance to judge errors, bias, or miscalibration?  Does system performance degrade if the judge is imperfect or biased?
- Why is a scalar judge score an appropriate mechanism for trajectory selection versus rank aggregation, uncertainty models, or majority-vote judges?
- Same backbone LLM is used for judge agent and verifier, no analysis of judge reliability or calibration


3. **“Holistic experience” concept lacks formal definition and mechanism explanation**

- The "holistic experience" terminology is used extensively but never formally defined, and transfer mechanism of such experience remains unspecified.
- Why does cross-task memory not induce negative transfer, especially across heterogeneous domains?  if tasks are heterogeneous, by what mechanism does experience remain reusable and beneficial?
- What type of knowledge actually transfers (Patterns? Metacognition? Strategies?)? Can the authors provide evidence or analysis.

4. **Multi-agent baseline coverage could be expanded**
Although many strong baselines are included, coverage of multi-agent reasoning systems could be broader.
- How does Xolver compare to debate-based approaches?
- How does it perform against multi-agent orchestration systems with richer communication?

**Questions:**

Refer to the section above in Weakness.

---

### Official Review · Reviewer_VBFG · 2025-11-01

**Soundness:** 3
**Presentation:** 2
**Contribution:** 3
**Rating:** 2
**Confidence:** 3

**Summary:**

This paper proposes Xolver, a training-free “generalist solver” that integrates multiple.
LLM-based agents (Planner, Dynamic Agents, Judge, Verifier) under Federated Agent Learning (FAL).
The key idea is that each agent independently solves the same problem without peer-to-peer communication; their reasoning traces are evaluated by a Judge agent and consolidated into a shared memory (DS), which is used for iterative refinement over several rounds. The paper claims that this design yields better generalization across math, coding, and multi-modal reasoning tasks (e.g., AIME, LiveCodeBench, GAIA).

**Strengths:**

1: The paper's primary strength is its outstanding performance. Xolver is evaluated on 14 diverse and challenging benchmarks spanning math, coding, and agentic tasks . It achieves new state-of-the-art results across all of them, including AIME (94.4) , LiveCodeBench (91.6) , HLE (57.3) and GAIA (90.1).

2: The dual-memory architecture is well-motivated. The intermediate shared memory (DS​) provides an effective mechanism for in-problem refinement, while the persistent episodic memory (DE​) in the Xolver (+) variant enables cross-problem learning, which is a key challenge in agent research


3: The paper provides a comprehensive set of ablations (Figure 2, Section 4, Appendix E) that clearly justify the framework's design. The ablations convincingly show the necessity of multi-iteration, multi-agent collaboration, the judge agent , and the benefits of external retrieval.


4: The four-agent pipeline (Planner → Dynamic Agents → Judge → Verifier) is well structured and intuitively appealing.

**Weaknesses:**

1. Insufficient Validation of the Judge Agent: The framework's success hinges on the Judge Agent's reliability, as it selects all experiences for the shared memory. However, the paper fails to validate the Judge's accuracy. The ablation study only proves the Judge is necessary (its removal degrades performance ), not that it is reliable. There is no comparison of the Judge's evaluations against ground-truth correctness, nor is there an error analysis for the Judge.


2. Unverified Reliability of the Judge on Long-Context Tasks (e.g., GAIA).
The paper assumes that GPT-4o can reliably assess correctness for long, multi-hop trajectories in GAIA-style tasks. However, no evidence or human validation is provided to confirm that the Judge can identify correct endpoints or reasoning consistency in such extended contexts. Given the absence of ground truth or truncation mechanisms, it is unclear how the Judge avoids overlooking late-stage reasoning errors or hallucinations in long outputs. This omission raises doubts about the robustness of the evaluation process.

	3. The paper emphasizes the importance of agent roles. However, it appears that only the [role] section of the prompt differs among agents. Besides, the reasoning content produced by each agent seems similar, showing no clear distinction in behavior or problem-solving perspective.

	4. Token usage issues: The report of the token usage part is not that detailed.  Only 5 of the benchmarks about the token usage are mentioned in Figure 12, which makes it hard for readers to assess the total cost and decide whether to use it.

	5. The paper’s framework for external memory retrieval is inconsistent and poorly defined across benchmarks. Section 2.2 describes D(E) as a static corpus of past experience. However, for over half of the evaluation benchmarks, they use self-retrieval or adaptive external retrieval with search engines, which are not explained in detail for their usage.

**Questions:**

1. For GAIA-style long-context reasoning trajectories, are the full trajectories directly passed to the Judge agent for scoring? How can the Judge reliably evaluate correctness without access to ground truth? Does the paper verify that GPT-4o can consistently detect correct endpoints in such settings?


2. The paper states that each agent is judged independently by the Judge agent. Why not evaluate multiple agent outputs jointly to capture diversity and consensus signals?


3. How can we be sure that the reported improvements come from the Judge-Memory architecture and not simply from iterative sampling as the results are compared with others which are at most pass@3 (i.e., running m agents for n rounds ≈ pass@mn)?


4.The paper claims strong performance across diverse benchmarks (AIME, LiveCodeBench, GAIA, etc.) under a unified system. What specific mechanism enables this generalization. Is it the shared memory design, the judge feedback, or task-specific prompt engineering? Please clarify which components are universal and which are tuned per benchmark.


5. Different roles of the Agent look to solve the problem in a similar way. Have you tried one type of the agents three or more times in one iteration and compared the differences?


6. Could you please clarify the episodic memory clearly, including external retrieval, self-retrieval, and adaptive external retrieval?


7. What kinds of problems can be solved at last if the first two rounds' answers are wrong? Why can the agent system correct the answer at last?

8. The general idea is similar to https://arxiv.org/pdf/2507.06229, which also introduces a memory mechanism for experience retrieval. However, this paper is not cited in the related work.

---

### Note · Authors · 2025-11-21

**Comment:**

We believe that we need to refine some concepts in this paper.

**Withdrawal Confirmation:**

I have read and agree with the venue's withdrawal policy on behalf of myself and my co-authors.